# Noise promotes independent control of gamma oscillations and grid firing within recurrent attractor networks

**Lukas Solanka[1,2,3], Mark CW van Rossum[2], Matthew F Nolan[1]\***

[1]Centre for Integrative Physiology, University of Edinburgh, Edinburgh, United Kingdom; [2]Institute for Adaptive and Neural Computation, Edinburgh, United Kingdom; [3]Neuroinformatics Doctoral Training Centre, School of Informatics, University of Edinburgh, Edinburgh, United Kingdom

**Abstract** Neural computations underlying cognitive functions require calibration of the strength of excitatory and inhibitory synaptic connections and are associated with modulation of gamma frequency oscillations in network activity. However, principles relating gamma oscillations, synaptic strength and circuit computations are unclear. We address this in attractor network models that account for grid firing and theta-nested gamma oscillations in the medial entorhinal cortex. We show that moderate intrinsic noise massively increases the range of synaptic strengths supporting gamma oscillations and grid computation. With moderate noise, variation in excitatory or inhibitory synaptic strength tunes the amplitude and frequency of gamma activity without disrupting grid firing. This beneficial role for noise results from disruption of epileptic-like network states. Thus, moderate noise promotes independent control of multiplexed firing rate- and gamma-based computational mechanisms. Our results have implications for tuning of normal circuit function and for disorders associated with changes in gamma oscillations and synaptic strength.

**\*For correspondence:**
mattnolan@ed.ac.uk

**Competing interests:** The authors declare that no competing interests exist.

## Introduction

Cognitive processes are mediated by computations in neural circuits and are often associated with gamma frequency oscillations in circuit activity. Gamma activity and cognitive performance often co-vary within tasks and between individuals, while cognitive deficits in psychiatric disorders such as autism and schizophrenia are linked to altered gamma frequency network dynamics (*Uhlhaas and Singer, 2012*; *Spellman and Gordon, 2014*). Such disorders are also linked to changes in the efficacy of excitatory glutamatergic and inhibitory GABAergic synapses (*Rubenstein and Merzenich, 2003*; *Lewis et al., 2012*). A critical and unresolved issue is the mechanistic relationship between gamma oscillations, the strength of excitation and inhibition, and circuit computations. On the one hand, neural codes based on firing rates may be sufficient for circuit computations (*Shadlen and Newsome, 1994*; *Histed and Maunsell, 2014*). In this scenario gamma oscillations might index circuit activation, but would not be required for computation. Evidence that rate coded computations and gamma oscillations arise from shared circuit mechanisms could be interpreted to support this view (*Lundqvist et al., 2010*; *Pastoll et al., 2013*), which predicts that when synaptic properties of a circuit are altered then gamma activity and the output of the rate-coded computation will co-vary. Alternatively, gamma oscillations, while sharing cellular substrates with rate-coded computations, may nevertheless support independent or multiplexed computational modes. For example, according to the communication through coherence hypothesis, tuning of gamma frequency activity may facilitate selective interactions between distant brain regions (*Fries, 2009*). In this scenario independent control of rate coded computation and gamma activity would be beneficial, for example by allowing tuning of

**eLife digest** When electrodes are placed on the scalp, or lowered into the brain itself, rhythmic waves of electrical activity are seen that reflect the coordinated firing of large numbers of neurons. The pattern of the waves varies between different brain regions, and according to what the animal or person is doing. During sleep and quiet wakefulness, slower brain waves predominate, whereas faster waves called gamma oscillations emerge during cognition—the act of processing knowledge.

Gamma waves can be readily detected in a region of the brain called the medial entorhinal cortex (MEC). This brain region is also known for its role in forming the spatial memories that allow an individual to remember how to navigate around an area they have previously visited. Individual MEC cells increase their firing rates whenever an individual is at specific locations. When these locations are plotted in two dimensions, they form a hexagonal grid: this 'grid cell map' enables the animal to keep track of its position as it navigates through its environment.

To determine how MEC neurons can simultaneously encode spatial locations and generate the gamma waves implicated in cognition, Solanka et al. have used supercomputing to simulate the activity of more than 1.5 million connections between MEC cells. Changing the strength of these connections had different effects on the ability of the MEC to produce gamma waves or spatial maps. However, adjusting the model to include random fluctuations in neuronal firing, or 'noise', was beneficial for both types of output. This is partly because noise prevented neuronal firing from becoming excessively synchronized, which would otherwise have caused seizures.

Although noise is generally regarded as disruptive, the results of Solanka et al. suggest that it helps the MEC to perform its two distinct roles. Specifically, the presence of noise enables relatively small changes in the strength of the connections between neurons to alter gamma waves—and thus affect cognition—without disrupting the neurons' ability to encode spatial locations. Given that noise reduces the likelihood of seizures, the results also raise the possibility that introducing noise into the brain in a controlled way could have therapeutic benefits for individuals with epilepsy.

coherence without disrupting multiplexed rate-coded computations. However, it is unclear how this could be achieved in circuits where gamma and rate-coded computations share common synaptic mechanisms, as this would require variation in synaptic properties to differentially affect gamma activity and the rate coded computation.

We address these issues using a model that accounts, through a common synaptic mechanism, for gamma oscillations and spatial computation by neurons in layer 2 of the medial entorhinal cortex (MEC) (*Pastoll et al., 2013*). The rate-coded firing of grid cells in the MEC is a well-studied feature of neural circuits for spatial cognition (*Moser and Moser, 2013*). During exploration of an environment individual grid cells are active at multiple locations that together follow a hexagonal grid-like organization. At the same time MEC circuits generate periods of activity in the high gamma frequency range (60–120 Hz) nested within a slower theta (8–12 Hz) frequency network oscillation (*Chrobak and Buzsáki, 1998*). Analysis of spatial correlations in grid firing, of manipulations to grid circuits, and recording of grid cell membrane potential in behaving animals, collectively point towards continuous two-dimensional network attractor states as explanations for grid firing (*Bonnevie et al., 2013*; *Domnisoru et al., 2013*; *Schmidt-Hieber and Häusser, 2013*; *Yoon et al., 2013*). In layer II of the MEC, which has the highest known density of grid cells (*Sargolini et al., 2006*), stellate cells that project to the dentate gyrus of the hippocampus are the major population of excitatory neurons (*Gatome et al., 2010*). These excitatory (E) neurons do not appear to influence one another directly but instead interact via intermediate inhibitory (I) neurons (*Dhillon and Jones, 2000*; *Couey et al., 2013*; *Pastoll et al., 2013*). Models that explicitly incorporate this recurrent E-I-E connectivity can account for grid firing through velocity-dependent update of network attractor states (*Pastoll et al., 2013*). When these models are implemented with excitable spiking neurons they also account for theta-nested gamma frequency network oscillations (*Pastoll et al., 2013*). The influence in these, or other classes of attractor network models, of the strength of E to I or I to E connections on gamma oscillations and grid firing, or other attractor computations, has not been systematically investigated.

We find that while gamma oscillations and grid firing are both sensitive to the strength of excitatory and inhibitory connections, their relationship differs. Although their underlying synaptic

substrates are identical, gamma activity nevertheless provides little information about grid firing or the presence of underlying network attractor states. Thus, gamma activity is not a good predictor of rate-coded computation. Unexpectedly, we find the range of E- and I- synaptic strengths that support gamma and grid firing is massively increased by moderate intrinsic noise through a mechanism involving suppression of seizure-like events. In the presence of moderate noise differences in synaptic strength can tune the amplitude and frequency of gamma across a wide range with little effect on grid firing. We obtain similar results in implementations of E-I models in which connectivity is probabilistic and in models extended to include additional I to I and E to E connections. Our results suggest constraints for extrapolation of differences in gamma activity to mechanisms for cognition, identify noise as a critical factor for successful circuit computation, and suggest that tuning of excitatory or inhibitory synaptic strength could be used to control gamma-dependent processes multiplexed within circuits carrying out rate coded computations.

## Results

To systematically explore relationships between strengths of excitatory and inhibitory synapses, computations and gamma activity, we initially take advantage of models that account for both grid firing and theta-nested gamma oscillations through E-I-E interactions (*Pastoll et al., 2013*). In these models a layer of E cells sends synaptic connections to a layer of I cells, which in turn feedback onto the E cell layer (*Figure 1A*). For attractor dynamics to emerge the strength of E and I connections are set to depend on the relative locations of neurons in network space (*Figure 1B*). While suitable connectivity could arise during development through spike timing-dependent synaptic plasticity (*Widloski and Fiete, 2014*), here the connection profiles are fixed (*Pastoll et al., 2013*). To vary the strength of excitatory or inhibitory connections in the network as a whole we scale the strength of all connections relative to a maximum conductance value ($g_E$ or $g_I$ for excitation and inhibition respectively) (*Figure 1B*). We also consider networks in which the connection probability, rather than

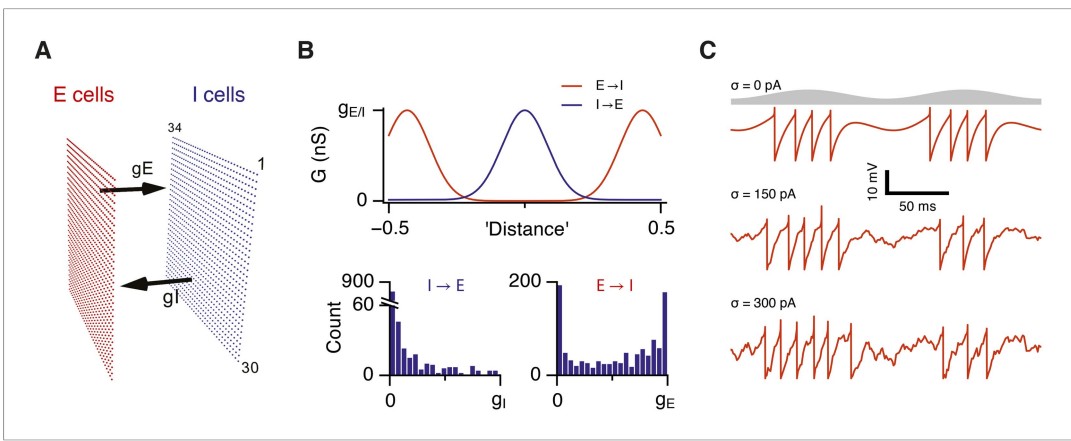

**Figure 1**. Attractor network model with feedback inhibition and theta frequency inputs. (**A**) A schematic of populations of excitatory cells (E cells, red) and inhibitory cells (I cells, blue) on a twisted torus of size 34 × 30 neurons. The synaptic coupling between the two populations was parameterized by the inter-population peak synaptic conductances $g_E$ (E → I synapses) and $g_I$ (I → E synapses). (**B**) Top: Plots illustrate peak synaptic conductances of E (red) and I (blue) synapses as a function of the distance between pre- and post-synaptic neurons. Bottom: Distributions of synaptic weights from all I cells onto an E cell in the model (left) and from all E cells onto an I-cell (right). Parameters $g_I$ and $g_E$ determine maximal values of these distributions. (**C**) Examples of the membrane potential of an isolated E cell during two consecutive theta cycles in networks without noise (white noise input current standard deviation σ = 0 pA), with an intermediate amount of noise (σ = 150 pA) and with noise levels doubled (σ = 300 pA). Theta signal is illustrated in grey.

The following figure supplement is available for figure 1:

**Figure supplement 1**. Synaptic weights in scaled and probabilistic variants of the network.

its strength, varies according to the relative position of neurons in the network (*Figure 1—figure supplement 1*). Each E and I cell is implemented as an exponential integrate and fire neuron and so its membrane potential approximates the dynamics of a real neuron, as opposed to models in which synaptic input directly updates a spike rate parameter. Addition of noise to a single E or I cell increases variability in its membrane potential trajectory approximating that seen in vivo (*Figure 1C*) (*Domnisoru et al., 2013*; *Pastoll et al., 2013*; *Schmidt-Hieber and Häusser, 2013*). Given that all neurons in the model are implemented as exponential integrate-and-fire neurons and that in total the model contains >1.5 million synaptic connections, we optimized a version of the model to enable relatively fast simulation and automated extraction and analysis of generated data (see 'Materials and methods'). In this way the effect on grid firing of $31 \times 31$ combinations of $g_E$ and $g_I$ could be evaluated typically using >50 nodes on a computer cluster in approximately 1 week.

## Intrinsic noise increases the range of synaptic strengths that support grid firing

What happens to grid firing patterns when the strengths of excitatory and/or inhibitory synaptic connections in the model are modified? To address this we first evaluated grid firing while simulating exploration within a circular environment with a network from which noise sources were absent (*Figure 2A*). When we reduce the strength of connections from I cells by threefold and increase the strength of connections from E cells by threefold we find that grid firing is abolished (*Figure 2Ab* vs *Figure 2Aa*). Exploring the parameter space of $g_E$ and $g_I$ more systematically reveals a relatively restricted region that supports grid firing (*Figure 2D* and *Supplementary file 1A–D*). Rather than the required $g_I$ and $g_E$ being proportional to one another, this region is shifted towards low values of $g_I$ and high $g_E$. Thus, the ability of recurrently connected networks to generate grid fields requires specific tuning of synaptic connection strengths.

Because neural activity in the brain is noisy (*Shadlen and Newsome, 1994*; *Faisal et al., 2008*), we wanted to know if the ability of the circuit to compute location is affected by noise intrinsic to each neuron (*Figure 1C*). Given that continuous attractor networks are often highly sensitive to noise (*Zhang, 1996*; *Eliasmith, 2005*), we expected that intrinsic noise would reduce the parameter space in which computation is successful. In contrast, when we added noise with standard deviation of 150 pA to the intrinsic dynamics of each neuron, we found that both configurations from *Figure 2Aa,b* now supported grid firing patterns (*Figure 2Ba,b*). When we considered the full space of E and I synaptic strengths in the presence of this moderate noise we now found a much larger region that supports grid firing (*Figure 2E* and *Supplementary file 1E–H*). This region has a crescent-like shape, with arms of relatively high $g_I$ and low $g_E$, and low $g_I$ and high $g_E$. Thus, while tuning of $g_I$ and $g_E$ continues to be required for grid firing, moderate noise massively increases the range of $g_E$ and $g_I$ over which grid fields are generated.

When intrinsic noise was increased further, to 300 pA, the parameter space that supports grid firing was reduced in line with our initial expectations (*Figure 2Ca,b,F* and *Supplementary file 1I–L*). To systematically explore the range of $g_E$ and $g_I$ over which the network is most sensitive to the beneficial effects of noise we subtracted grid scores for simulations with 150 pA noise from scores with deterministic simulations (*Figure 2G*). This revealed that the unexpected beneficial effect of noise was primarily in the region of the parameter space where recurrent inhibition was strong. In this region, increasing noise above a threshold led to high grid scores, while further increases in noise progressively impaired grid firing (*Figure 2H*). In probabilistically connected networks, the range of $g_E$ and $g_I$ supporting grid firing was reduced, but the shape of the parameter space and dependence on noise was similar to the standard networks (*Figure 2—figure supplement 1*), indicating that the dependence of grid firing on $g_E$ and $g_I$, and the effects of noise, are independent of the detailed implementation of the E-I attractor networks.

How closely does the firing of I cells in the simulated networks correspond to inhibitory activity in behaving animals, and to what extent is the pattern of I cell firing affected by $g_E$, $g_I$ and noise? While there is little data on the spatial firing of interneurons in the MEC, recent evidence indicates that the majority of parvalbumin positive interneurons have firing fields with significant spatial stability, but low spatial sparsity and grid scores compared to excitatory grid cells (*Buetfering et al., 2014*). A possible interpretation of these data is that parvalbumin positive cells are unlikely to fulfill the roles of I cells predicted in E-I models. However, in networks that we evaluate here in which E cells have grid firing

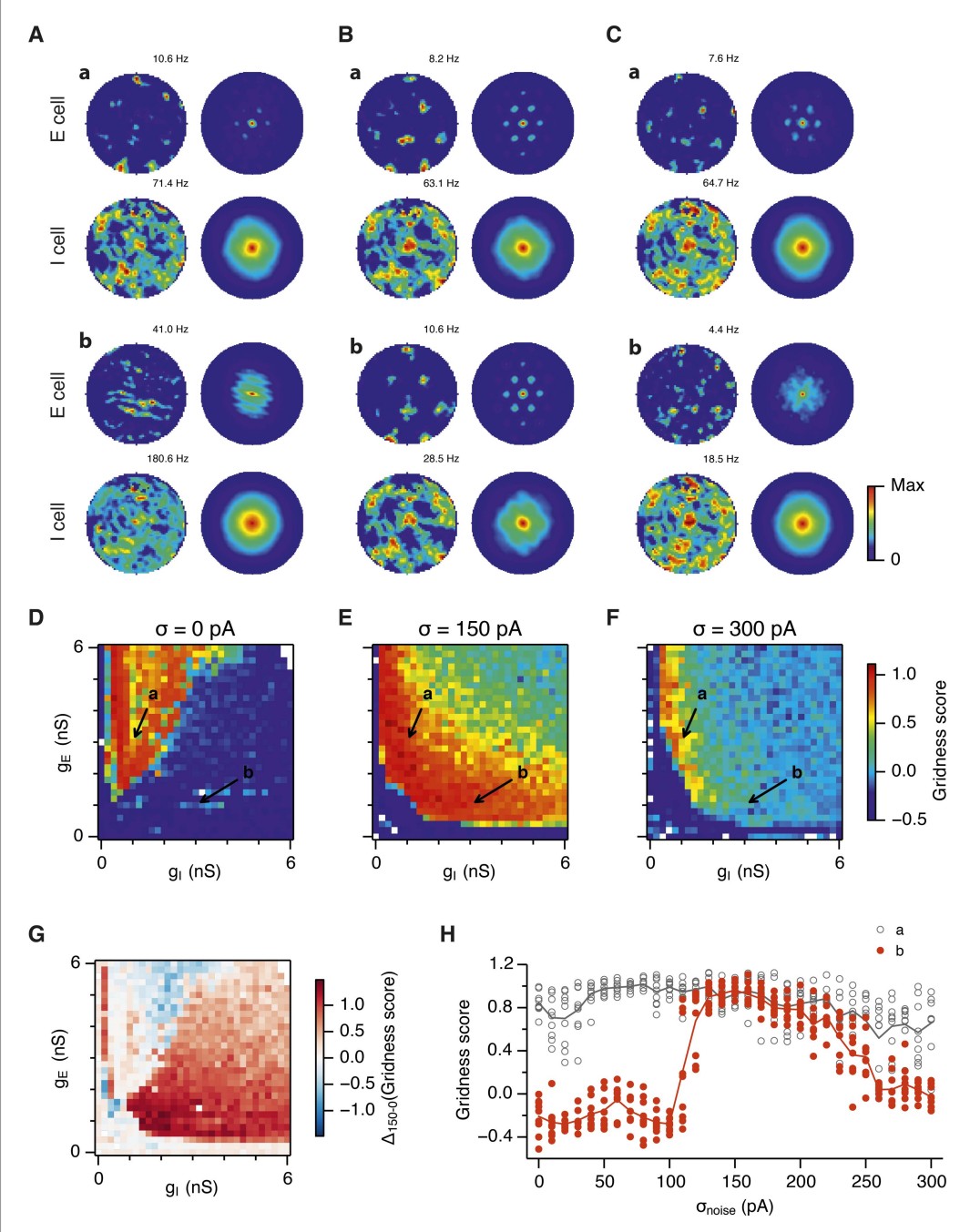

**Figure 2**. Noise increases the range of synaptic strengths that support grid firing. (**A–C**) Example spatial firing fields (left) and spatial autocorrelation plots (right) of E and I cells for networks without noise (**A**; σ = 0 pA), with noise level set to σ = 150 pA (**B**), and noise level set to σ = 300 pA (**C**) and with the strengths of recurrent synaptic connections indicated by arrows in (**D–F**). Maximal firing rate is indicated to the top right of each spatial firing plot. The range of spatial autocorrelations is normalized between 0 and 1. (**D–F**) Gridness score as a function of $g_E$ and $g_I$ for networks with each noise level. Each item in the color plot is an average gridness score of four simulation runs. Arrows indicate the positions of grid field and autocorrelation examples from simulations illustrated in (**A–C**). Simulations that did not finish in a specified time interval (5 hr) are indicated by white color. (**G**) Difference between gridness scores of networks with σ = 150 pA and networks with σ = 0 pA plotted as a function of $g_E$ and $g_I$. (**H**) Gridness score plotted as a function of the standard deviation of intrinsic noise. Each noise level comprises simulations from a neighborhood of $g_E$ and $g_I$ surrounding a center point in the parameter space (center included) indicated by arrows in (**D–F**).

*Figure 2. continued on next page*

*Figure 2. Continued*

The following figure supplements are available for figure 2:

**Figure supplement 1**. Sensitivity of grid firing to changes in feedback inhibition, excitation and noise levels in networks with connection probability between pairs of neurons drawn according to the synaptic profile functions in *Figure 1B*.

**Figure supplement 2**. Spatial information and sparsity of firing fields of E and I cells.

**Figure supplement 3**. Gridness scores of I cells.

**Figure supplement 4**. Spatial firing fields in networks with uncorrelated spatial input applied to each I cell.

fields in the presence of moderate noise, I cell firing fields also have a much lower spatial information content and spatial sparsity than the corresponding E cell firing fields (E cells: spatial sparsity $0.788 \pm 0.061$, spatial information: $1.749 \pm 0.32$ bits/spike; I cells: spatial sparsity $0.239 \pm 0.018$, spatial information $0.243 \pm 0.024$ bits/spike; $p < 10^{-16}$ for comparisons of both spatial sparsity and information; paired t-test; data range is indicated as mean $\pm$ standard deviation) (*Figure 2A–C* and *Figure 2—figure supplement 2*). Spatial autocorrelograms of simulated I cell firing fields also do not contain the six hexagonally organized peaks that are characteristic of grid fields (*Figure 2A–C*). Nevertheless, I cell spatial autocorrelograms produce positive grid scores ($0.39 \pm 0.16$; *Figure 2—figure supplement 3*), although these are reduced compared to scores for the E cells in the same networks (E cells: $0.796 \pm 0.157$; $p < 10^{-16}$; paired t-test; mean $\pm$ SD) and in many networks are below the threshold considered previously to qualify as grid like (cf. Figure 4B of *Buetfering et al., 2014).* When we evaluated the dependence of I cell spatial firing on $g_E$, $g_I$ and noise, it appeared to be similar to that of E cells (*Figure 2—figure supplement 3*). To assess whether grid scores of I cells can be reduced further in E-I networks while maintaining grid firing by E cells, we investigated networks in which uncorrelated spatial input is applied to each I cell (*Figure 2—figure supplement 4*). In these simulations E cells had grid scores of $0.57 \pm 0.25$, spatial sparsity of $0.78 \pm 0.03$ and spatial information of $1.69 \pm 0.18$ bits/spike, whereas I cells had grid scores of $0.16 \pm 0.2$ ($p < 10^{-16}$, paired t-test), spatial sparsity of $0.21 \pm 0.01$ ($p < 10^{-16}$, paired t-test) and spatial information of $0.2 \pm 0.01$ bits/spike ($p < 10^{-16}$, paired t-test; range of all data sets is mean $\pm$ SD). Thus, spatial firing of I cells has a similar dependence on noise, $g_E$ and $g_I$ to grid cells, conventional indices of spatial firing are nevertheless much lower for I cells in E-I networks compared to E cells, and grid firing by E cells in E-I networks is relatively robust to disruption of the rotational symmetry of I cell firing fields.

Together these simulations demonstrate that attractor circuit computations that generate grid firing fields require specific tuning of $g_E$ and $g_I$. In the absence of noise grid firing is supported in relatively restricted regions of parameter space. Optimal levels of noise, which produce single cell membrane potential fluctuations of a similar amplitude to experimental observations (*Domnisoru et al., 2013*; *Pastoll et al., 2013*; *Schmidt-Hieber and Häusser, 2013*), promote grid firing by reducing the sensitivity of grid computations to the strength of recurrent synaptic connections, particularly when inhibition is relatively strong and excitation is weak.

## Differential sensitivity of gamma oscillations and grid firing to the strength of E and I synapses

Is the sensitivity of gamma frequency oscillations to synaptic strength and to noise similar to that of grid firing? To evaluate gamma activity we recorded synaptic currents from single E and I cells across multiple theta cycles (*Figure 3A–C*). For the network configurations illustrated in *Figure 2Aa,b* and in which intrinsic noise is absent, we observed synaptic events entrained to theta cycles (*Figure 3Aa,b*). However, the timing and amplitude of synaptic events typically differed between theta cycles and no consistent gamma rhythm was apparent. In contrast, in the presence of noise with standard deviation 150 pA we observed nested gamma frequency synaptic activity with timing that was consistent between theta cycles (*Figure 3Ba*). In this condition the frequency of the gamma oscillations was reduced and their amplitude increased by raising $g_I$ and lowering $g_E$ (*Figure 3Bb*). With a further

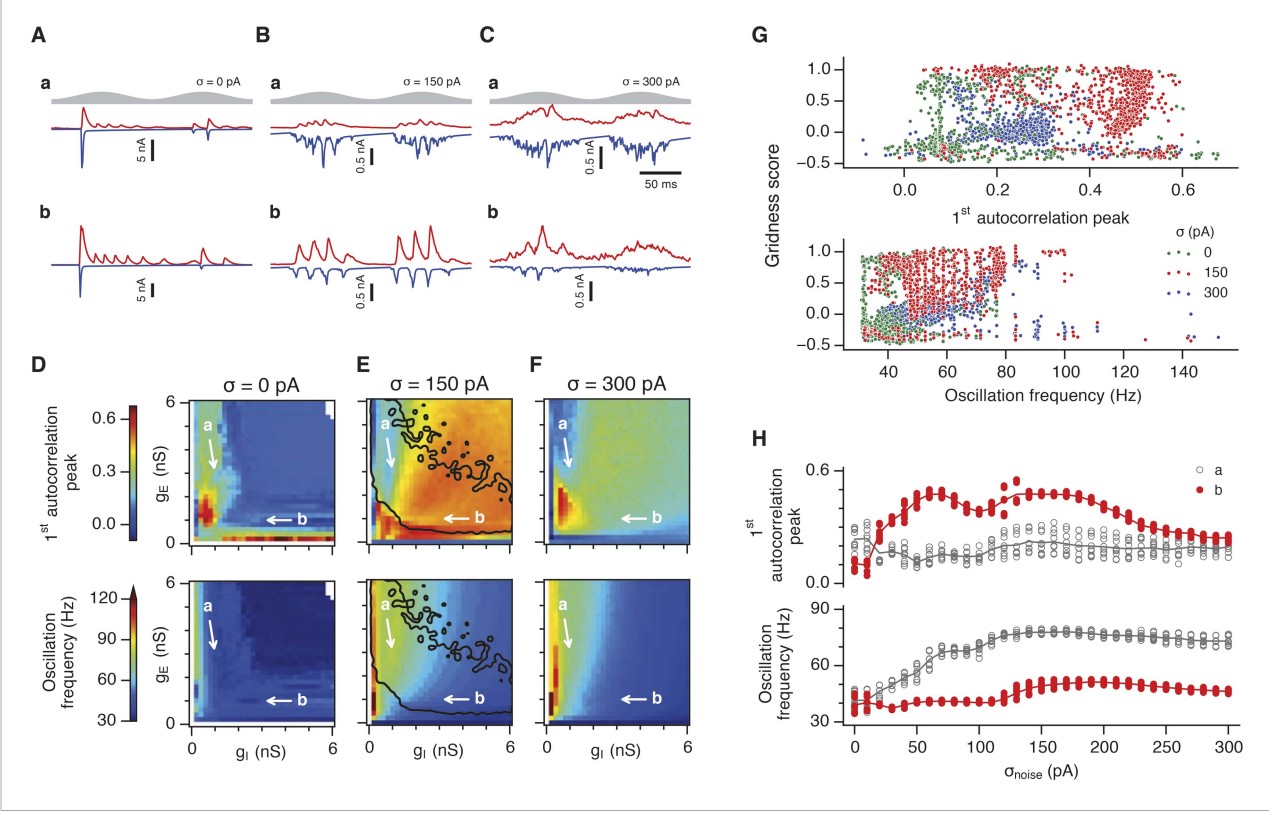

**Figure 3**. Differential sensitivity of gamma oscillations and grid fields to changes in the strength of E and I synapses. (**A–C**) Examples of inhibitory (red) and excitatory (blue) synaptic currents recorded respectively from excitatory and inhibitory neurons from simulations highlighted by arrows in panels (**D–F**). (**D–F**) Top: Correlation value at the first local maximum of the autocorrelation of inhibitory synaptic currents (I → E cells, 25 randomly selected E cells), plotted as a function of $g_E$ and $g_I$, for networks without noise (**D**), with noise level set to σ = 150 pA (**E**), and noise level set to σ = 300 pA (**F**). Each point is an average over five simulation trials. In these simulations velocity and place cell inputs were disabled. The duration of simulations was 10 s. Bottom: Frequency corresponding to the peaks of the autocorrelation functions for simulations in the top panels. Black lines in (**E**) indicate the region from *Figure 2E* where the gridness score = 0.5. (**G**) Scatter plots show gridness score as a function of gamma oscillation strength (top) and frequency (bottom) for simulations with noise absent (green), with an intermediate level of noise (red) and highest simulated noise level (blue). Each dot represents data from a single network configuration. (**H**) Top: Gamma oscillation strength plotted as a function of standard deviation of the noise current. Grey color indicates simulations with $g_E$ = 3 nS, $g_I$ = 1 nS (**A**). Red color indicates simulations with $g_E$ = 1 nS, $g_I$ = 3 nS (**B**). Bottom: Frequency corresponding to the detected autocorrelation peak.

The following figure supplements are available for figure 3:

**Figure supplement 1**. Sensitivity of gamma oscillations to changes in the strength of E and I synapses in networks with connection probability between pairs of neurons drawn according to the synaptic profile functions in *Figure 1B*.

**Figure supplement 2**. Scatter plots of gridness score as a function of the amplitude of gamma oscillations.

**Figure supplement 3**. Scatter plots of gridness score as a function of the detected oscillation frequency.

**Figure supplement 4**. Amplitude and frequency of gamma oscillations in the $g_E$ and $g_I$ parameter regions where grid fields are robust.

increase in noise to 300 pA, gamma activity remained entrained to theta cycles, but became less ordered (*Figure 3Ca,b*).

To explore gamma activity across a wider range of $g_I$ and $g_E$ we automated quantification of the strength and frequency of oscillatory input to E cells (see 'Materials and methods'). In the absence of noise gamma frequency activity only occurred for a narrow range of $g_I$ and $g_E$ (*Figure 3D*). Strikingly, following addition of moderate noise the region of parameter space that supports gamma activity was

massively expanded (*Figure 3E*). Within this space, the amplitude of gamma increased with increasing inhibition, whereas the frequency was reduced. As noise is increased further the amplitude and frequency of gamma oscillations are reduced (*Figure 3F*). We found a similar dependence of gamma oscillations on noise, $g_E$ and $g_I$ in networks with probabilistic connectivity (*Figure 3—figure supplement 1*). Thus, intrinsic noise modifies the amplitude and frequency of nested gamma oscillations.

To determine whether there is a systematic relationship between values of $g_E$ and $g_I$ that generate gamma and grid firing we compared the gridness score and gamma scores for each circuit configuration (*Figure 3G*, *Figure 3—figure supplements 2, 3*). We found this relationship to be complex and highly sensitive to noise. However, we did not find any evidence for strong linear relationships between gamma amplitude or gamma frequency and grid score ($R^2 < 0.12$ for all comparisons), while gamma amplitude and frequency provided only modest amounts of information about grid scores ($0.27 < MIC < 0.33$ and $0.27 < MIC < 0.37$ respectively). The relationship between noise intensity and gamma differed from that for grid computations. Whereas, grids emerged above a sharp noise threshold (*Figure 2H*), for the same regions in parameter space the frequency and amplitude of gamma oscillations varied smoothly as a function of noise (*Figure 3H*). Thus, neither the frequency nor the power of gamma appears to be a good predictor of grid firing.

When we considered only regions of parameter space that generate robust grid fields (grid score >0.5), we found circuits generating almost the complete observed range of gamma amplitudes (0.02 < autocorrelation peak < 0.59) and frequencies (31 Hz < frequency < 102 Hz) (*Figure 3—figure supplement 4*). For example, considering the crescent shaped region of E-I space that supports grid firing in the presence of intermediate noise (the region within the isocline in *Figure 3E*), when $g_I$ is high and $g_E$ low then the amplitude of gamma is relatively low and the frequency high. Moving towards the region where $g_I$ is high and $g_E$ is low, the amplitude of gamma is increased and the frequency is reduced. Thus, variation of synaptic strength across this region of E-I space can be used to tune the properties of gamma activity while maintaining the ability of the network to generate grid fields.

Together these data indicate that an optimal level of noise promotes emergence of gamma oscillations, while the properties of oscillations may depend on the relative strength of synaptic connections. The relationship between gamma and synaptic strength differs to that for grid computations. Strikingly, while gamma activity provides relatively little information about grid firing, differential sensitivity of gamma and grid firing to $g_E$ and $g_I$ provides a mechanism for circuits to tune gamma frequency activity while maintaining the ability to compute rate coded grid firing fields.

## Noise promotes attractor computation by opposing seizures

Given the emergence of a large parameter space that supports grid firing following introduction of moderate noise, we were interested to understand how noise influences the dynamics of the E-I circuits. One possibility is that in networks that fail to generate grid firing fields network attractor states form, but their activity bumps are unable to track movement. In this scenario disrupted grid firing would reflect incorrect control of network activity by velocity signals. Alternatively, deficits in grid firing may reflect failure of network attractor states to emerge. To distinguish these possibilities we investigated formation of activity bumps in network space over the first 10 s following initialization of each network (*Figure 4*).

Our analysis suggests that the deficit in grid firing in deterministic compared to noisy networks reflects a failure of attractor states to emerge. For deterministic simulation of the points in parameter space considered in *Figure 2Aa*, which are able to generate grid patterns, we found that a single stable bump of activity emerged over the first 2.5 s of simulated time (*Figure 4Aa*). In contrast, for deterministic simulation of the point considered in 2Ab, which in deterministic simulations did not generate grid patterns, a single stable bump fails to emerge (*Figure 4Ab*). Quantification across the wider space of $g_E$ and $g_I$ values (see 'Materials and methods') indicated that when $g_I$ is low there is a high probability of bump formation as well as grid firing, whereas when $g_I$ is high the probability of both is reduced (*Figure 4B*). In contrast to the deterministic condition, for circuits with intrinsically noisy neurons activity bumps emerged in the first 1.25 s following initialization of the network (*Figure 4Ac–e*) and the area of parameter space that supported bump formation was much larger than that supporting grid firing (*Figure 4B*). Plotting gridness scores as a function of bump probability

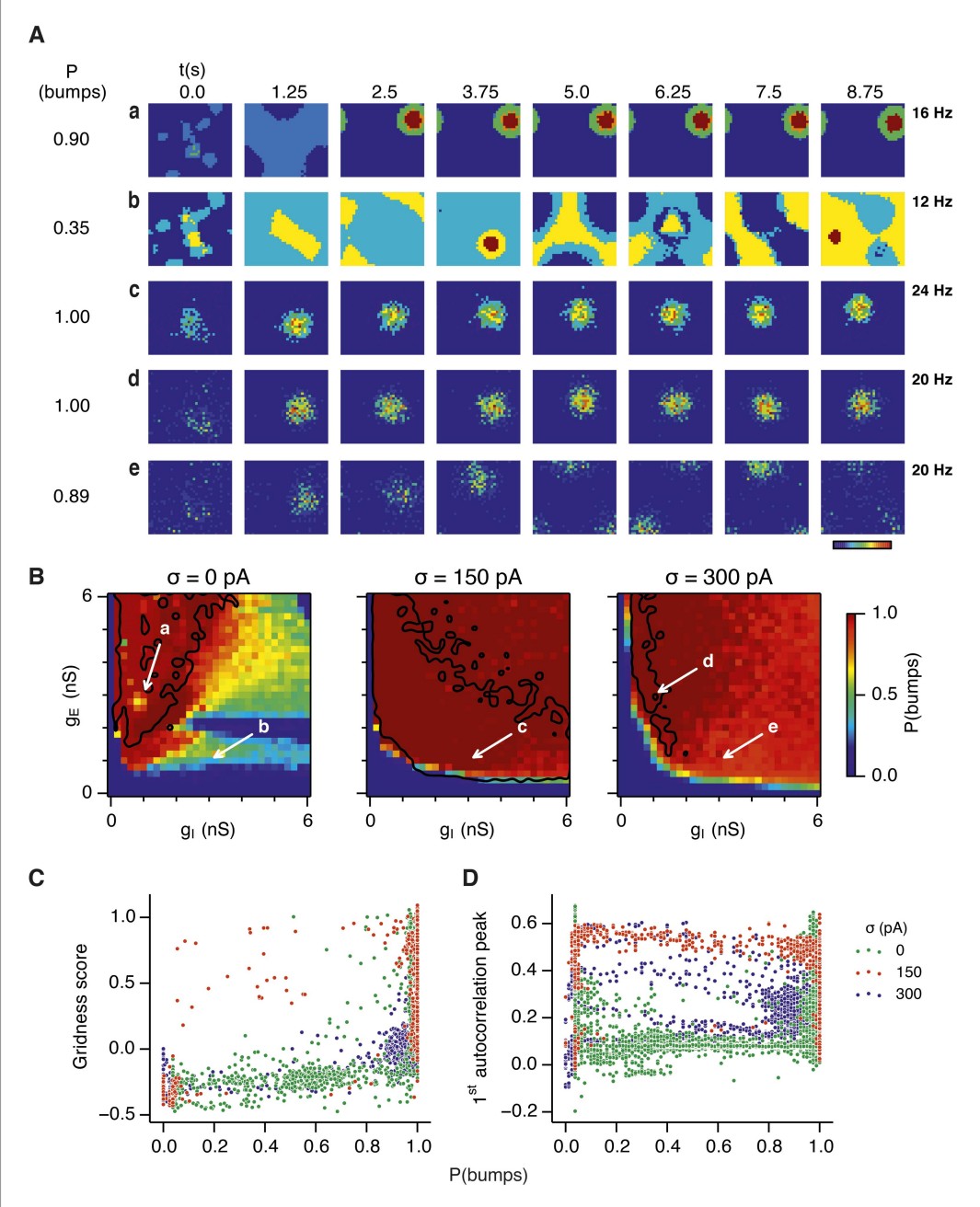

**Figure 4**. Noise promotes formation of continuous attractors. (**A**) Examples of snapshots of network activity of E cells from simulations in which velocity and place cell inputs are inactivated. Each row shows a simulation trial with a value of $g_E$ and $g_I$ highlighted by an arrow in panel (**B**). The corresponding probability of bump formation (P(bumps)) and the maximal firing rate is indicated to the left and right, respectively. (**B**) Color plots show probability of bump formation (P(bumps)), for the simulated range of $g_E$ and $g_I$ and the three simulated noise levels. Each color point is an average of five 10 s simulation runs. Arrows show positions in the parameter space of examples in (**A**). Black lines indicate the regions where the gridness score = 0.5 (cf. **Figure 2D–F**). (**C**) Relationship between gridness score computed from the grid field simulation runs (**Figure 2D–F**) and the probability of bump formation (**B**). (**D**) Relationship between gamma oscillation strength (**Figure 3D–F**) and the probability of bump formation (**B**). Each color in (**C** and **D**) represents one noise level and each dot in the scatter plots corresponds to simulations of a single pair of values of $g_E$ and $g_I$.

The following figure supplement is available for figure 4:

**Figure supplement 1**. Sensitivity of bump attractor spontaneous drift to variations in $g_E$ and $g_I$ and noise levels.

indicated that bump formation was necessary, although not sufficient for grid formation (*Figure 4C*), while plotting the first autocorrelation peak as a function of bump probability supported our conclusion that grid computation and gamma activity are not closely related (*Figure 4D*). Together, these data indicate that noise promotes formation of attractor bumps in network activity and in deterministic simulations the failure of the circuit to generate attractor states largely accounts for disrupted grid firing.

In noisy networks the presence of low grid scores for networks with high bump scores (*Figure 4C*) is explained by sensitivity of these network configurations to noise-induced drift. This is illustrated by the region of parameter space from *Figure 2Ab*, where $g_I$ is relatively high and $g_E$ relatively low, and which in deterministic simulations fails to generate bumps or grids. With moderate noise, this point generates bumps that show little drift (*Figure 4Ac*), whereas as noise is increased further the bump begins to drift (*Figure 4Ae*). In contrast, at the point illustrated in *Figure 2Aa*, which forms grids and bumps in the presence or absence of noise, activity bumps are relatively stable in each condition (*Figure 4Aa,d*), although drift increases with greater noise (*Figure 4—figure supplement 1*). Thus, intrinsic noise has two opposing effects on bump formation. For much of the parameter space we consider moderate noise promotes emergence of bumps and grids, while across all of parameter space noise reduces bump stability leading to deterioration of grids.

To investigate how addition of noise promotes emergence of network attractor states we investigated the dynamics of neurons in the simulated circuits. We focus initially on the point in parameter space identified in *Figure 2Ab*, where grids are found in the presence of moderate noise, and bumps are found when noise is moderate or high. When we examined times of action potentials generated by all neurons in this circuit, we find that in the absence of noise the network generates hyper-synchronous seizure-like states at the start of each theta cycle (*Figure 5A* and *Figure 5—figure supplement 1A*). The number of E cells active on each theta cycle differs, but their activity is typically restricted to the rising phase of theta, and there is no consistent structure in the pattern of activated neurons. The number of simultaneously active I cells is also greatest at the start of each theta cycle. The I-cells continue to fire over the theta cycle, but their synchronization declines. When moderate noise is added to the circuit only a subset of E-cells are active on each theta cycle, forming an activity bump (*Figure 5B* and *Figure 5—figure supplement 1B*). The I-cells are active at gamma frequency and the formation of an activity bump in the E-cell population is reflected by an inverted bump in the I-cell population activity (*Figure 5B*). With increased noise there is a similar overall pattern of activity, but spike timing becomes more variable, causing the bumps to drift and reducing the degree of synchronization at gamma frequencies (*Figure 5C* and *Figure 5—figure supplement 1C*).

To determine whether these changes in network dynamics are seen across wider regions of parameter space we first quantified the presence of seizure like events from the maximum population firing rate in any 2 ms window over 10 s of simulation time (E-rate$_{max}$). Strikingly, we found that in the absence of noise epochs with highly synchronized activity were found for almost all combinations of $g_E$ and $g_I$, whereas these seizure-like events were absent in simulations where noise was present (*Figure 5D*). Interestingly, while grids emerge in deterministic networks in regions of E-I space where E-rate$_{max}$ is relatively low, there is a substantial region of parameter space in which E-rate$_{max}$ is >400 Hz, but grids are nevertheless formed. It is possible that seizure-like states may be rare in this region of parameter space and so do not interfere sufficiently with attractor dynamics to prevent grid firing. To test this we calculated for each combination of $g_E$ and $g_I$ the proportion of theta cycles having events with population-average rate >300 Hz ($P_{E\text{-rate}} > 300$). For values of $g_E$ and $g_I$ where grid fields are present $P_{E\text{-rate}} > 300$ was relatively low, indicating that seizure-like events are indeed rare (*Figure 5E*). Consistent with this, when we plotted grid score as a function of $P_{E\text{-rate}} > 300$, we found that $P_{E\text{-rate}} > 300$ was relatively informative about the gridness score in networks without noise (MIC = 0.624) and a low value of $P_{E\text{-rate}} > 300$ was necessary for grid firing (*Figure 5F*). In contrast, E-rate$_{max}$ was less informative of grid firing ($0.392 \leq$ MIC $\leq 0.532$) and a wide range of values were consistent with grid firing (*Figure 5F*). Thus, while grid firing is compatible with occasional seizure-like events, when seizure-like events occur on the majority of theta cycles then grid firing is prevented.

Because seizure-like events tend to initiate early on the depolarizing phase of each theta cycle, we asked if synchronization by theta frequency drive plays a role in their initiation. When theta frequency input was replaced with a constant input with the same mean amplitude, the power of gamma oscillations was still dependent on the levels of noise and changes in $g_E$ and $g_I$ (*Figure 6—figure supplement 1*). However, in contrast to simulations with theta frequency input (*Figure 5D,E*),

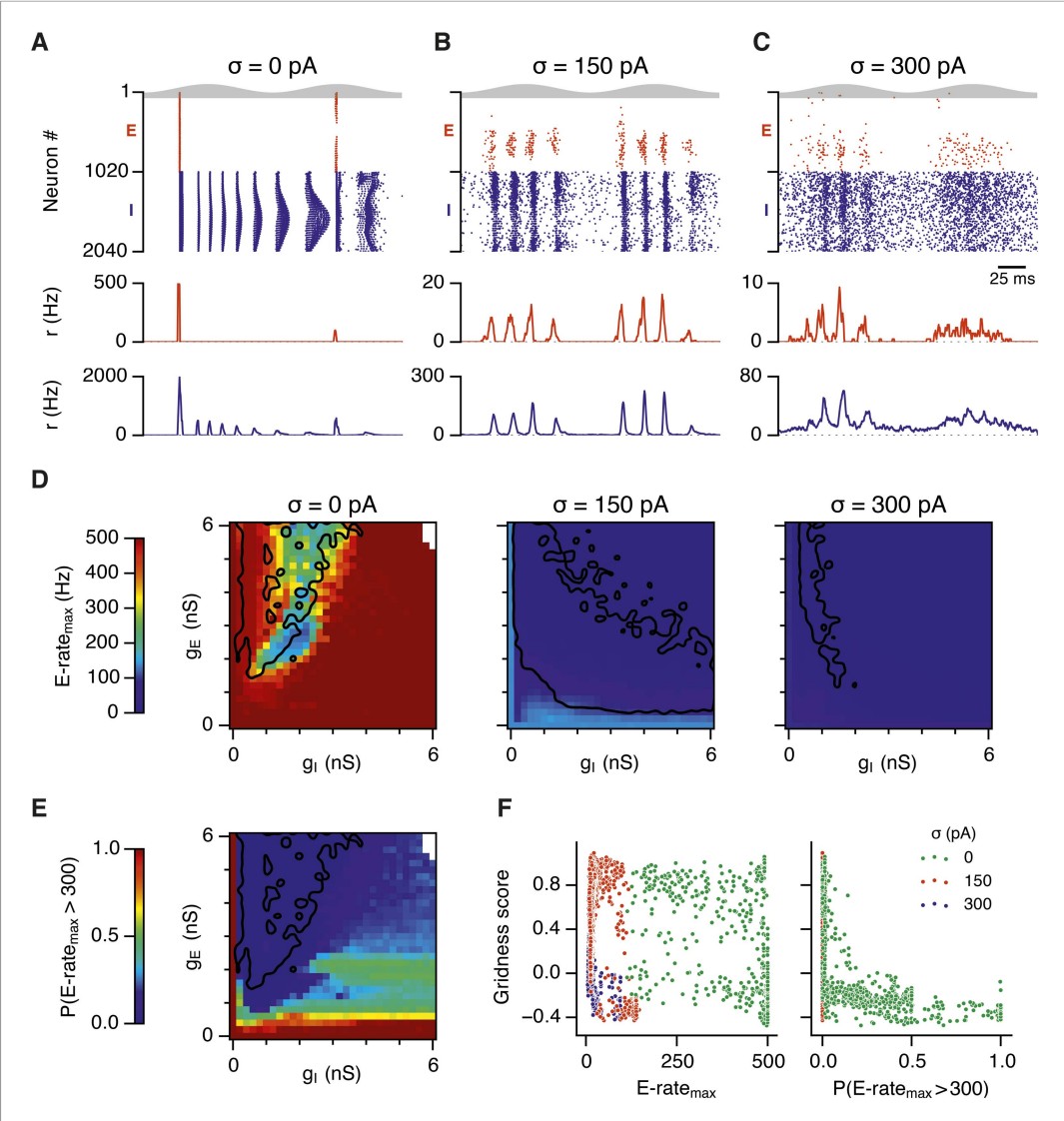

**Figure 5.** Noise opposes generation of seizure-like states. (**A–C**) Raster plots show activity of all neurons in the excitatory (red) and inhibitory (blue) populations for the duration of two theta cycles (top), along with the average population firing rates for both populations (center and bottom; calculated with a sliding rectangular window with 2 ms duration and 0.5 ms time step), for networks where noise is absent (**A**; σ = 0), with noise set to σ = 150 pA (**B**), and with noise set to σ = 300 pA (**C**). Simulations were performed in the absence of animal movement and place cell input; $g_E$ = 1 nS and $g_I$ = 3 nS. (**D**) Maximal average population firing rate of E cells estimated from the whole simulation run (10 s; 500 ms at the beginning of the simulation excluded) for each simulated level of noise. Each point is an average of maxima from five simulation runs. (**E**) Probability of the maximal population-average firing rate during each theta cycle exceeding 300 Hz, that is, at least 60% of E cells firing synchronously within a time period of 2 ms in the parameter space of $g_E$ and $g_I$ when σ = 0 pA. Black lines indicate regions where gridness score equals 0.5. (**F**) Scatter plots show the relationship between gridness score and the maximal firing rate during the simulation (left) and the probability of the maximal population-average firing rate during each theta cycle exceeding 300 Hz (right).
The following figure supplement is available for figure 5:

**Figure supplement 1.** Examples of activity in the network.

noise-free networks without theta exhibited hyper-synchronous firing only when $g_E$ was <0.5 nS (**Figure 6A**) and generated grid firing fields almost in the complete range of $g_E$ and $g_I$ (**Figure 6D,G**). Addition of noise in the absence of theta had mostly detrimental effects on grid firing (**Figure 6E,F,H,I**

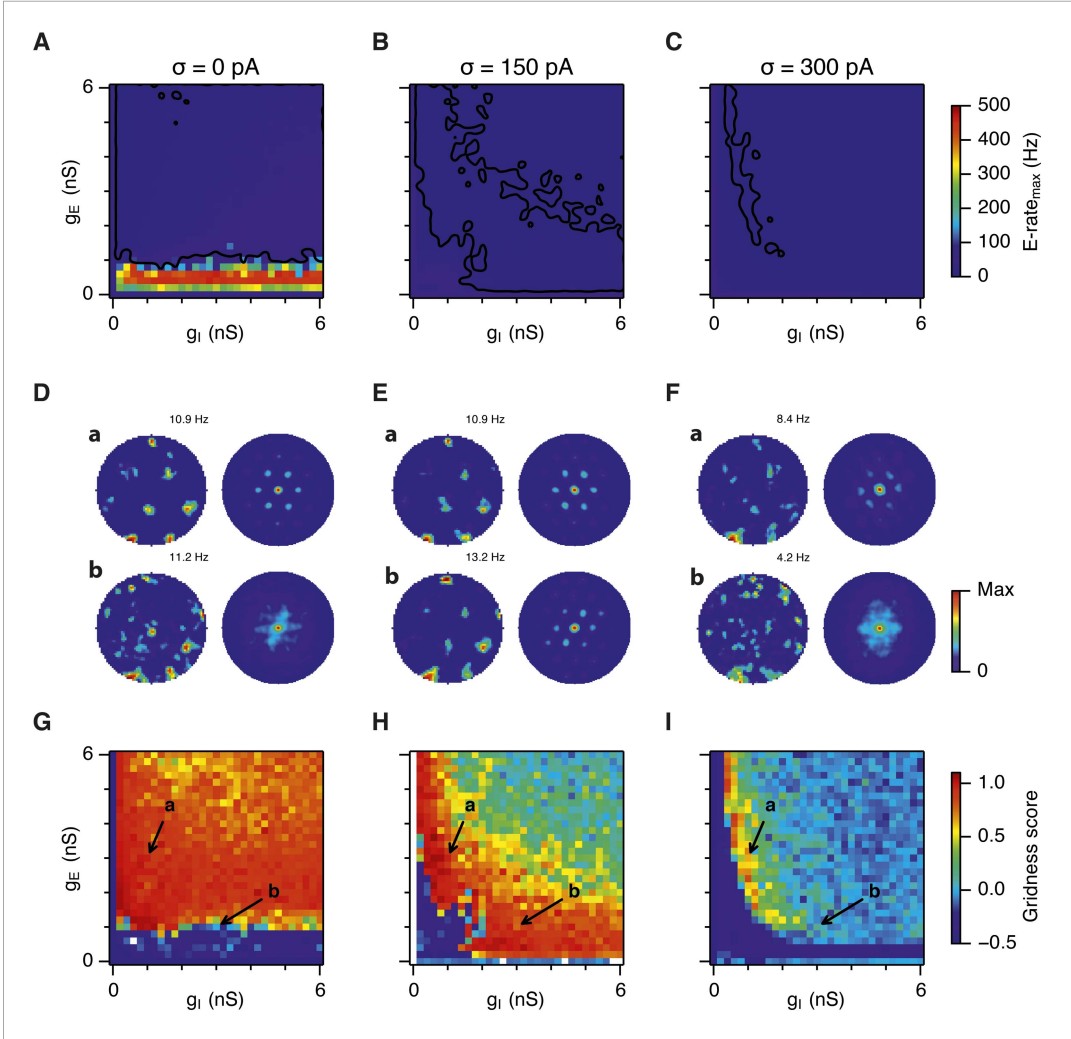

**Figure 6**. Seizure-like states and grid firing fields in networks without theta frequency inputs. (**A–C**) Maximal average population firing rate of E cells estimated from the whole simulation run (10 s; 500 ms at the beginning of the simulation excluded) for each simulated level of noise indicated by σ, in networks with theta frequency inputs replaced with a constant input with the same mean amplitude. Each point is an average of maxima from five simulation trials. Black lines indicate the regions from (**G–H**) where gridness score = 0.5. (**D–F**) Example spatial firing fields (left) and autocorrelation plots (right) for the specific values of $g_E$ and $g_I$ indicated by arrows in (**G–I**), corresponding to the three simulated noise levels. Maximal firing rate is indicated at the top right of each spatial firing plot. The range of spatial autocorrelations is normalized between 0 and 1. (**G–I**) Gridness score as a function of $g_E$ and $g_I$, for each simulated level of noise. Each item in the color plot is an average gridness score of three simulation runs of 600 s duration. Arrows indicate the positions of grid field and autocorrelation examples from simulations illustrated in (**D–F**). Simulations that did not finish in a specified time interval (5 hr) are indicated by white color.

The following figure supplements are available for figure 6:

**Figure supplement 1**. Effect of replacing theta frequency inputs by a constant input with an equal mean amplitude.

**Figure supplement 2**. Effect of noise on gridness scores in networks without theta frequency inputs.

**Figure supplement 3**. Firing rates of E cells.

**Figure supplement 4**. Calibration of the gain of the velocity inputs.

**Figure supplement 5**. Effectivity of the place cell resetting mechanism as a function of $g_E$ and $g_I$ and noise levels.

and *Figure 6—figure supplement 2*). Interestingly, with intermediate levels of noise, the subregion with high gridness scores (>0.5) retained its crescent-like shape (*Figure 6E,H*), but was smaller when compared to the networks with theta frequency inputs (size of regions with and without theta: 488/961 vs 438/961), while the range of gamma frequencies present was much lower than in networks containing theta drive. Together, these data indicate that moderate noise prevents emergence of seizure like states by disrupting synchronization of the attractor network by the shared theta frequency drive. In networks with moderate noise theta drive promotes grid firing and enables a wide range of gamma frequencies to be generated without disrupting grid firing.

Our analysis points towards suppression of seizure-like events as the mechanism by which moderate noise promotes grid firing, while interactions between noise and theta appear important for the capacity to multiplex grid firing with a wide range of gamma frequencies. However, we wanted to know if other factors might contribute to these beneficial roles of noise. Grid fields may also fail to form if overall activity levels are too low, in which case neurons with grid fields instead encode head direction (*Bonnevie et al., 2013*). This observation is unlikely to explain our results as the mean firing rate of E cells in networks that generated grid firing fields (grid score >0.5, networks with $g_E$ or $g_I$ set to 0 excluded) was in fact lower than the firing rate of networks without grid fields (1.2; 1.0; 1.0 Hz grid fields vs 3.0; 2.7; 1.2 Hz no grid fields, in networks with $\sigma = 0$; 150; 300 pA respectively). There was also no systematic relationship between grid score and firing frequency (*Figure 6—figure supplement 3*). We also wanted to know if other properties of grid fields vary as a function of $g_E$ and $g_I$. Parameters used to calibrate velocity integration by the grid network varied very little with changes in $g_E$ and $g_I$ (*Figure 6—figure supplement 4*), whereas drift increased with $g_I$ (*Figure 4—figure supplement 1*) and place cell input was most effective in opposing attractor drift in noisy networks with high gridness scores (*Figure 6—figure supplement 5*). These data are consistent with suppression of seizure like events as the mechanism by which noise promotes grid firing, while interactions between noise and theta frequency inputs profoundly influence the dynamics of attractor networks that generate grid fields.

## Recurrent inhibition increases the frequency of gamma activity and promotes grid firing

Our analysis so far focuses on E-I attractor networks as simple models of grid firing that are compatible with the finding that synaptic interactions between stellate cells in layer 2 of the MEC are mediated via inhibitory interneurons (*Dhillon and Jones, 2000*; *Couey et al., 2013*; *Pastoll et al., 2013*). However, there is evidence that interneurons active during theta-nested gamma activity make connections to one another as well as to stellate cells (*Pastoll et al., 2013*). To establish whether this recurrent inhibition substantially modifies our conclusions from simpler E-I networks, we extended the E-I model to also include synapses between interneurons (see 'Materials and methods'). In the resulting E-I-I networks, in the absence of noise, grid firing emerges across a much larger region of parameter space compared to E-I networks (*Figure 7A*, *Figure 7—figure supplements 1–4*). However, as in E-I networks occasional seizure like activity was present across a wide range of $g_E$ and $g_I$ (*Figure 7—figure supplement 5*), and gamma frequency activity was largely absent (*Figure 7D,G*). Following addition of noise with standard deviation of 150 pA to E-I-I networks, grid firing was maintained, seizure like activity was abolished, and gamma like activity emerged (*Figure 7B,E,H* and *Figure 7—figure supplement 5*). Increasing the noise amplitude to 300 pA reduced grid firing and interfered with the emergence of gamma oscillations (*Figure 7C,F,I* and *Figure 7—figure supplements 1–5*). Importantly, just as in E-I networks, the presence of moderate noise in E-I-I networks enables tuning of gamma activity by varying $g_E$ and $g_I$ while maintaining the ability of the networks to generate grid firing fields. Gamma activity had a higher frequency in E-I-I compared to E-I networks, with a greater proportion of the parameter space supporting gamma frequencies >80 Hz. This higher frequency gamma is similar to fast gamma observed experimentally in the MEC (cf. *Chrobak and Buzsáki, 1998*; *Colgin et al., 2009*; *Pastoll et al., 2013*). Thus, by including additional features of local circuits in layer 2 of the MEC, E-I-I models may more closely recapitulate experimental observations. Nevertheless, E-I-I networks maintain the ability, in the presence of moderate noise, for variation in $g_E$ and $g_I$ to tune gamma oscillations without interfering with grid firing.

Finally, we asked if addition of synaptic connections between excitatory cells modifies the relationship between gamma, noise, $g_E$ and $g_I$. While the E-I model is consistent with the connectivity

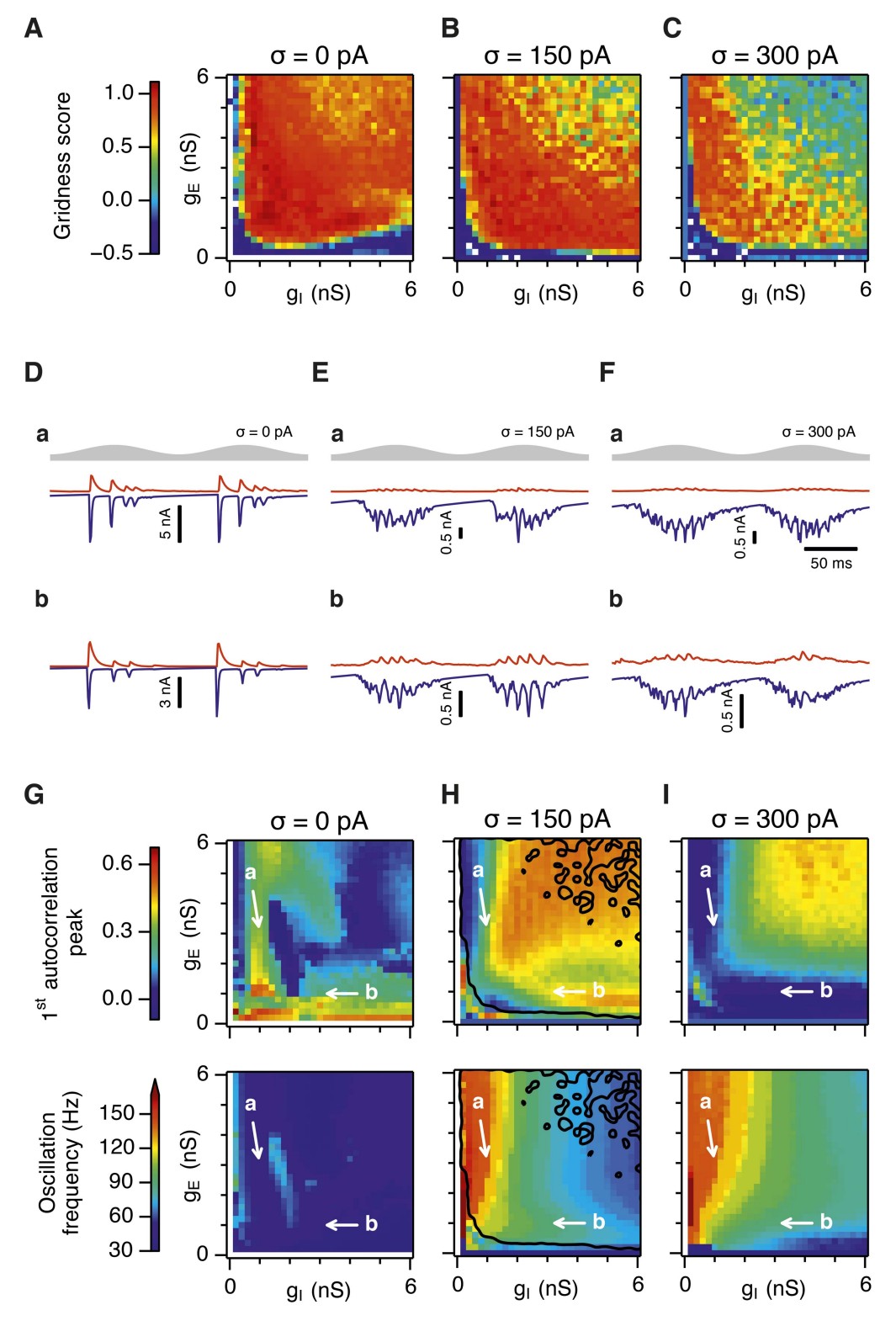

**Figure 7**. Gridness scores and gamma activity in networks with recurrent inhibition. (**A–C**) Gridness score as a function of $g_E$ and $g_I$ for networks without noise (**A**; σ = 0 pA), with noise level set to σ = 150 pA (**B**), and noise level set to σ = 300 pA (**C**). Simulations that did not finish in a specified time interval (5 hr) are indicated by white color. (**D–F**) Examples of inhibitory (red) and excitatory (blue) synaptic currents recorded respectively from excitatory and

*Figure 7. continued on next page*

*Figure 7. Continued*

inhibitory neurons from simulations highlighted by arrows in panels (**G–I**). (**G–I**) Top: Correlation value at the first local maximum of an autocorrelation of inhibitory synaptic currents (I → E cells, 25 randomly selected E cells), plotted as a function of $g_E$ and $g_I$, for networks without noise (**G**), with noise level set to σ = 150 pA (**H**), and noise level set to σ = 300 pA (**I**). Each point is an average over five simulation trials. In these simulations velocity and place cell inputs were disabled. The duration of simulations was 10 s. Bottom: Frequency corresponding to the peaks of the autocorrelation functions for simulations in the top panels. Black lines in (**H**) indicate the regions from (**B**) where gridness score = 0.5.

The following figure supplements are available for figure 7:

**Figure supplement 1**. Spatial firing fields in networks that contain recurrent I → I synapses.

**Figure supplement 2**. Continuous attractors in networks that contain direct I → I synapses.

**Figure supplement 3**. Sensitivity of bump attractor spontaneous drift to variations in $g_E$, $g_I$ and noise levels in networks that contain direct I → I synapses.

**Figure supplement 4**. Calibration of the gain of the velocity inputs in networks that contain direct I → I synapses.

**Figure supplement 5**. Seizure-like states in networks that contain direct I → I synapses.

**Figure supplement 6**. Sensitivity of grid firing to changes in inhibition and excitation in networks that contain direct E → E synapses.

**Figure supplement 7**. Sensitivity of gamma oscillations to changes in inhibition and excitation in networks that contain direct E → E synapses.

**Figure supplement 8**. Continuous attractors in networks that contain direct E → E synapses.

**Figure supplement 9**. Seizure-like states in networks that contain direct E → E synapses.

**Figure supplement 10**. Probability of bump formation and network activity plots in networks with structured E → E and unstructured E → I and I → E connections.

between stellate cells in layer 2 of the MEC, adjacent pyramidal cells may also have grid firing properties. Unlike stellate cells, pyramidal cells interact with one another directly via excitatory connections and indirectly via inhibitory interneurons (*Couey et al., 2013*). To assess the impact of E-E connections, we first extended the E-I model to allow each E cell to excite other E cells that are nearby in neuron space. The dependence of grid firing, gamma oscillations, and bump formation on noise, $g_E$ and $g_I$ was similar to E-I networks (*Figure 7—figure supplements 6–9*). We also attempted to evaluate networks in which E-E connections were structured, but E-I and I-E connections were uniformly distributed. However, in these networks we were unable to identify parameters that support formation of stable activity bumps (*Figure 7—figure supplement 10*). This is consistent with instability of simpler network attractors based on E-E connections (*Seung et al., 2000*).

## Discussion

We investigated the relationship between rate coded spatial computations and nested gamma oscillations in attractor network models of grid firing. While in the models we consider rate coding and gamma oscillations share the same neural substrate, that is projections from a population of E cells to an I cell population, which in turn projects back to the E cell population, we find that their sensitivity to variations in excitatory and inhibitory synaptic strengths nevertheless differs. A moderate level of noise promotes generation of both grid fields and nested gamma oscillations, primarily by the disruption of epileptic-like firing of E and I cells in the network. When the strength of E or I connections is varied in the presence of moderate noise a wide range of gamma frequency and power

can be obtained without affected grid firing. Thus, noise can be beneficial for computations performed by the nervous system, while the frequency and power of multiplexed gamma oscillations can be tuned independently of rate-coded grid computations, suggesting a mechanism for differential control of multiplexed neural codes.

Our results suggest a novel beneficial role for noise. In general noise in the nervous system is believed to distort the fidelity of transmitted signals (*Faisal et al., 2008*). Exceptions are stochastic resonance phenomena in which noise promotes detection of small amplitude signals by individual neurons (*Longtin et al., 1991*; *Benzi et al., 1999*; *Shu et al., 2003*), improvements in signal coding through desynchronization of neuronal populations (*Hunsberger et al., 2014*) and emergence of stochastic weak synchronization in interneuron networks (*Tiesinga and Jose, 2000*). The beneficial role for noise that we identify here differs from these phenomena in that it emerges through interactions between populations of neurons and because the grid cell attractor network performs a computation—generation of a spatial code from velocity inputs—rather than propagating input signals. We find that by opposing emergence of hyper-sychronous seizure-like states noise allows the network to generate stable bump attractor states. Noise prevents the seizure-like states by desynchronizing neuronal responses to common theta input. We were able to identify this role for noise because spiking and synaptic dynamics are explicitly represented in the simulated network. These dynamics are absent from other attractor network models of grid firing (*Fuhs and Touretzky, 2006*; *Guanella et al., 2007*; *Burak and Fiete, 2009*). They are also absent from other models of theta-nested gamma oscillations that simulate two-dimensional dynamical systems of E and I populations with theta modulated inputs to the network (*Onslow et al., 2014*). Thus, intrinsic cellular and synaptic dynamics in conjunction with noise sources may be important in accounting for computations and oscillatory activity in neural networks.

The distinct control of rate coded grid computations and gamma oscillations by noise, $g_E$ and $g_I$ was independent of the detailed implementation of the E-I models we considered and was maintained in more complex models incorporating I-I and E-E coupling. Current available experimental data appears to be insufficient to distinguish between these different models. For example, our analysis of interneuron firing indicates that while E-I models predict that interneurons will have spatial firing fields, they have lower spatial information content, spatial sparsity and grid scores than E cells and therefore may be difficult to detect in existing experimental datasets and with current analysis tools. Thus, evidence previously interpreted to argue against E-I based mechanisms for grid firing may in fact not distinguish these from other possible mechanisms. Indeed, we found that grid firing by E cells can be maintained during spatial input that distorts the spatial firing pattern of I cells (*Figure 2—figure supplement 4*). While these simulations establish in principle that E-I based attractor networks can generate grid outputs even when spatial firing of many E and I cells in the network is not clearly grid-like, the extent to which these networks can account for additional details of experimental observations, for example weak periodic patterns in the spatial autocorrelation of the firing fields of some PV interneurons (cf. *Buetfering et al., 2014*, *Figure 4a*), is not yet clear. Our results are consistent with local synaptic connections, in addition to those between E cells and I cells, having important functional roles. For example addition of synapses between interneurons to E-I networks causes an overall increase in the frequency of gamma activity and in the stability of grid firing. Nevertheless, we find that in these modified networks moderate noise still enables variation in $g_E$ and $g_I$ to tune gamma oscillations independently from grid firing.

An intriguing aspect of our results is that they suggest novel approaches to suppressing seizures and to promoting normal cognitive function. Seizures have previously been suggested to result from deficits in inhibition or from alterations in intrinsic excitability of neurons (*Lerche et al., 2001*; *Treiman, 2001*). We show that seizures can be induced when these properties are held constant simply by reducing levels of noise within a circuit. A future experimental challenge for dissecting the contribution of intrinsic noise to seizures will be to target biological noise sources. In the brain noise arises from ion channel gating and from background synaptic activity. It is therefore difficult to manipulate noise sources without also affecting intrinsic excitability or excitation-inhibition balance. However, it may be feasible to add noise to circuits through transcranial magnetic stimulation (*Ruzzoli et al., 2010*). In this case our simulations predict that addition of noise may restore epileptic circuits to normal activity. This mechanism may explain why focal electrical stimulation of the entorhinal cortex in patients with seizures leads to an enhancement of memory performance (*Suthana et al., 2012*).

While correlations between gamma oscillations and various cognitive and pathological brain states are well established, the proposed computational roles of gamma oscillations have been difficult to reconcile with rate-coded representations with which they co-exist. We were able to address this issue directly by analyzing a circuit in which gamma oscillations and rate-coded computations arise from a shared mechanism. Rather than gamma serving as an index of rate-coded computation, we find instead that there is a substantial parameter space across which rate-coded computation is stable, while the amplitude and frequency of theta-nested gamma oscillations varies. Our analysis leads to several new and testable predictions. First, tuning of recurrent synaptic connections could be used to modify gamma oscillations without affecting rate-coded computation. If multiple networks of the kind we simulate here correspond to grid modules providing input to downstream neurons in the hippocampus (*Stensola et al., 2012*), then adjusting $g_E$ or $g_I$ would alter gamma frequency with minimal effect on the grid firing pattern of each module. If the downstream neurons integrate input at the gamma time scale, then this should lead to re-mapping of their place representation in the absence of any change in either the strength of their synaptic inputs or the information they receive from upstream grid cells. Adjustment of $g_E$ and $g_I$ could be achieved dynamically through actions of neuromodulators (*Marder, 2012*), or on slower developmental time scales (*Widloski and Fiete, 2014*). Second, subtle differences in gamma could be a sensitive index of network pathology at stages before deficits in rate coded computation are apparent. If cognitive deficits in psychiatric disorders reflect a failure of rate coded computation, then our analysis predicts that a change in noise within a circuit, in addition to synaptic modification, may be necessary for deficits to emerge. From this perspective it is intriguing that seizure phenotypes are often associated with disorders such as autism (*Deykin and MacMahon, 1979*). Alternatively, cognitive deficits may result from a failure to coordinate gamma frequency synchronization of circuits that converge on downstream targets. In this case we expect cognitive deficits to be phenocopied by manipulations that affect gamma frequency or power without influencing rate-coded computations (*Sigurdsson et al., 2010*; *Spellman and Gordon, 2014*).

In conclusion, our systematic exploration of three dimensions of parameter space ($g_E$, $g_I$ and intrinsic noise) illustrates the complexity of relationships between rate-coded computation, gamma frequency oscillations and underlying cellular and molecular mechanisms. Our results highlight the challenges in straightforward interpretation of experiments in which these parameters are correlated to one another, (cf. *Wang and Krystal, 2014*). While there are parallels to investigations of pace-making activity in invertebrate circuits (*Marder and Taylor, 2011*), which demonstrate that many parameter combinations can account for higher order behavior, there are also critical differences in that the models we describe account for multiplexing of rate-coded computation and oscillatory activity, while the number of neurons and connections in the simulated circuit is much larger. Future experimentation will be required to test our model predictions for unexpected beneficial roles of noise and for control of gamma oscillations independently from grid firing by modulating the strength of excitatory and inhibitory synaptic connections.

## Materials and methods

The model comprised a network of exponential integrate and fire neurons (*Fourcaud-Trocmé et al., 2003*) implemented as a custom-made module of the NEST simulator (*Gewaltig and Diesmann, 2007*). The network investigated in the majority of simulations (*Figures 1–6*) is modified from that in *Pastoll et al. (2013)* and consists of excitatory (E) and inhibitory (I) populations of neurons that were arranged on a twisted torus with dimensions of 34 × 30 neurons. In networks where connection strengths were generated probabilistically instead of in an all-to-all way, the synaptic weights from E to I cells and vice versa were constant, while the probability of connection between the pre- and post-synaptic neuron was drawn according to *Figure 1B*. In addition, some networks also included direct uniform recurrent inhibition between I cells (*Figure 7*; referred to as E-I-I networks) or direct structured recurrent excitation between E cells (*Figure 7—figure supplements 6–10*). When recurrent excitation was present, synaptic weights between E cells followed the connectivity profile in which the strongest connection was between cells that were close to each other in network space (*Figure 1B*) and the weights between E and I cells were generated either according to synaptic profiles from *Figure 1B* (*Figure 7—figure supplements 6–9*) or the E-I connectivity was uniform with a probability of connection of 0.1 (*Figure 7—figure supplement 10*). E and I cells also received the theta current drive which was the sum of a constant amplitude positive current and a current with sinusoidal waveform (8 Hz). The constant component of the drive was required to activate the circuit,

while the theta drive frequency was chosen to reflect the frequency of theta oscillations in behaving animals. The amplitude (cf. Appendix 1) was chosen to produce theta modulation of I cell firing similar to that observed in behaving animals (cf. *Chrobak and Buzsáki, 1998*) and ex-vivo models of theta-nested gamma activity (cf. *Pastoll et al., 2013*). In order to oppose drift of the activity bump in networks that simulated exploration of the arena E cells received input from cells with place-like firing fields simulated as Poisson spiking generators with their instantaneous firing rate modeled as a Gaussian function of the animal position. Full details of the connectivity and network parameters are in Appendix 1.

In all simulations the networks were parameterized by the standard deviation of noise ($\sigma$) injected independently into each E and I cell and by synaptic scaling parameters ($g_E$ and $g_I$). Noise was sampled from a Gaussian distribution with standard deviation either set to $\sigma = 0$, 150 or 300 pA, or alternatively in the range of 0–300 pA in steps of 10 pA (*Figures 2H, 3H*). The peaks of the synaptic profile functions (*Figure 1B*) were determined by the $g_E$ and $g_I$ parameters that appropriately scaled the maximal conductance values of the excitatory and inhibitory connections respectively.

Gridness scores were estimated by simulating exploration in a circular arena with a diameter of 180 cm. For each value of $g_E$ and $g_I$ the simulations consisted of two phases. In the first phase, animal movement with constant speed and direction (vertically from bottom to top) was simulated in order to calibrate the gain of the velocity input to achieve 60 cm spacing between grid fields in the network. In the second phase, the calibrated velocity input gains were used during a simulation of realistic animal movements with duration of 600 s (*Hafting et al., 2005*). Each simulation was repeated 1–4 times. For each trial, gridness score was then estimated from an E or I cell located at position (0, 0) on the twisted torus. In simulations where interneurons received uncorrelated spatial inputs (*Figure 2—figure supplement 4*), gridness scores were estimated from 100 randomly selected E and I cells on the twisted torus.

For the analysis of bump attractor properties and gamma oscillations a separate set of simulations were run. For each value of $g_E$, $g_I$ and noise level, there were five trials of 10 s duration during which the velocity and place cell inputs were deactivated. For each trial spiking activity of all cells was recorded. In addition, inhibitory synaptic currents of 25 randomly selected E cells were saved and used for further analysis.

The strength and frequency of gamma oscillations were estimated from the inhibitory synaptic currents recorded from E cells. The currents were first band-pass filtered between 20 and 200 Hz. For each trace, its autocorrelation function was computed and the first local maximum was detected using a peak detection algorithm which was based on calculating the points in the autocorrelation function where the first difference of the signal changed sign from positive to negative and thus approximated the points where the first derivative was zero and the second derivative was negative. The strength and frequency of gamma oscillations was estimated from the correlation value and lag at the position of the first local maximum respectively.

Properties of bump attractors were estimated by fitting symmetric Gaussian functions onto successive snapshots of firing rates of each cell in the E population. For each snapshot this procedure yielded the position of the bump center and its width. The probability of bump formation was then estimated as a proportion of population-activity snapshots that were classified as bump attractors, that is, those fitted Gaussian functions whose width did not exceed the shorter side of the twisted torus. Other properties of bump attractors were estimated by analyzing successive positions of the bump attractor centers. Action potential raster plots of E and I populations (*Figure 5A–C*, *Figure 5—figure supplement 1* and *Figure 7—figure supplement 10*) show neuron indices that are flattened in a row-wise manner with respect to the two-dimensional twisted torus. Data points with white color in *Figure 5D,E* and *Figure 5—figure supplement 1A* have been excluded from analysis since the maximal firing rate of E cells exceeded 500 Hz/2 ms window.

The calculation of the maximal information coefficient (MIC) for the relationship between gridness score, gamma and bump scores was estimated by applying the MIC measure using the minepy package (*Albanese et al., 2013*). Calculations of spatial information were carried out according to (*Skaggs et al., 1996*). Spatial sparsity was calculated by following the procedure outlined in (*Buetfering et al., 2014*). All other data analysis and simulations were performed in Python.

## Acknowledgements

We thank Hugh Pastoll, Lukas Fisher and Paolo Puggioni for useful discussions. This work has made use of resources provided by the Edinburgh Compute and Data Facility (ECDF; www.ecdf.ed.ac.uk), which has support from the eDIKT initiative (www.edikt.org.uk).

# Additional information

### Funding

| Funder | Grant reference | Author |
|---|---|---|
| Biotechnology and Biological Sciences Research Council (BBSRC) | BB/L010496/1 | Lukas Solanka, Matthew F Nolan |
| Biotechnology and Biological Sciences Research Council (BBSRC) | BB/F529254/1 | Lukas Solanka, Mark CW van Rossum |
| Engineering and Physical Sciences Research Council (EPSRC) | EP/F500385/1 | Lukas Solanka, Mark CW van Rossum |

The funders had no role in study design, data collection and analysis, decision to publish, or preparation of the manuscript.

### Author contributions

LS, Conception and design, Acquisition of data, Analysis and interpretation of data, Drafting or revising the article; MCWR, Conception and design, Analysis and interpretation of data; MFN, Conception and design, Analysis and interpretation of data, Drafting or revising the article

### Author ORCIDs

Matthew F Nolan, http://orcid.org/0000-0003-1062-6501

# Additional files

### Supplementary file

• Supplementary file 1. Examples of spatial firing fields. (A-L) Top: Gridness score in the parameter space of the E and I synaptic strength scaling parameters ($g_E$ and $g_I$ respectively). Bottom: Firing fields of a single cell obtained by simulating animal movement, in the parameter region highlighted by black rectangle in the parameter space plot. Above each firing field is the estimated gridness score (left) and maximal firing rate in the firing field (right). Blank (white) locations in the parameter space are simulations that did not finish in the pre-specified time limit (5 hr). Noise level used in each set of simulations is shown by $\sigma_{noise}$. Color scale in the firing field plots ranges from 0 Hz (dark blue) to the maximal firing rate for each of the firing fields (dark red).

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

## Appendix 1

## Supplementary methods

### Neuron membrane and synaptic dynamics

Each neuron's membrane potential ($V_m$) is governed by the passive membrane equation:

$$C_m \dot{V}_m = I_m + I_{syn} + I_{ext} + \eta, \tag{1}$$

in which the total membrane current is a sum of four separate components: the trans-membrane current ($I_m$), the total synaptic current ($I_{syn}$), the current injected externally from other brain regions ($I_{ext}$) and $\eta \sim \mathcal{N}(0, \sigma^2)$, which is the noise current with zero mean and appropriate standard deviation in the range of 0–300 pA.

For E cells, the trans-membrane current

$$I_m = g_L(E_L - V_m) + g_{AHP}(t)(E_{AHP} - V_m) + g_L \Delta_T \exp\left(\frac{V_m - V_T}{\Delta_T}\right), \tag{2}$$

contains the leak conductances ('L' subscript), after-spike hyperpolarisation conductance ('AHP' subscript) and an exponential part that initiates a spike when the membrane potential gets close to the threshold ($V_T$). After each spike, there is a reset of membrane potential and the AHP conductance:

$$V_m \to V_r,$$

$$g_{AHP} \to g_{AHP_{max}}. \tag{3}$$

The I cells do not possess an AHP, but instead contain a simple adaptation term. The trans-membrane current has the following form:

$$I_m = \left(g_L + g_{ad}(t)\right)(E_L - V_m) + g_L \Delta_T \exp\left(\frac{V_m - V_T}{\Delta_T}\right). \tag{4}$$

The $g_{ad}$ term adds an extra conductance after each spike, that is, after the spike:

$$V_m \to V_r,$$

$$g_{ad} \to g_{ad} + g_{ad_{inc}}. \tag{5}$$

We used adaptation for the I cells in order to include refractory properties after each spike. The frequency vs current (F-I) relationship of the standard leaky integrate-and-fire neuron model has a steep slope right after the firing threshold has been crossed. This is an undesirable property because a neuron's firing rate is overly sensitive to small current changes. To linearize the F-I curve we used adaptation. The adaptation was not specifically tuned to produce the current model behavior and other mechanisms could be used as well (e.g., after-spike hyperpolarization as was done in the case of E cells).

Both AHP and adaptation conductances ($g_{AHP}$ and $g_{ad}$ respectively) decay exponentially:

$$\dot{g}_{AHP} = -\frac{g_{AHP}}{\tau_{AHP}},$$

$$\dot{g}_{ad} = -\frac{g_{ad}}{\tau_{ad}}. \tag{6}$$

In **Equations 2, 4**, the term $\Delta_T$ is defined as the spike slope factor (**Fourcaud-Trocmé et al., 2003**) and it measures the sharpness of the spike initiation. The closer this parameter is to zero, the faster spike initiation will happen when $V_m$ gets close to $V_T$. For the exponential integrate

and fire neuron, in the limit $\Delta_T \to 0$, the model becomes equivalent to a leaky integrate and fire neuron (**Fourcaud-Trocmé et al., 2003**).

The synaptic current for each neuron is a sum of the AMPA, NMDA and GABA$_A$ synaptic currents collected from spikes of all other neurons:

$$I_{\text{syn}}(t) = g_{\text{GABA}_A}(t)(E_{\text{GABA}_A} - V_m) + g_{\text{AMPA}}(t)(E_{\text{AMPA}} - V_m) + g_{\text{NMDA}}(t)(E_{\text{NMDA}} - V_m). \qquad (7)$$

In networks that do not contain recurrent E $\to$ E connections we set $g_{\text{AMPA}} = g_{\text{NMDA}} = 0$ for the E cells, and $g_{\text{GABA}_A} = 0$ for I cells. In other network variants (with E $\to$ E or I $\to$ I connectivity) these synaptic strengths are non-zero. E $\to$ E, as well as E $\to$ I synapses thus both contain the NMDA component. Connections from place cells were modeled as AMPA conductances only (cf. description of place cell inputs). The synaptic conductances $g_{\text{AMPA}}$, $g_{\text{NMDA}}$ and $g_{\text{GABA}_A}$ of a postsynaptic neuron $i$ were modeled as exponentials with pre-defined time constants (see **Appendix table 1** for the parameter values):

$$\dot{g}^i_{\text{AMPA}} = -\frac{g_{\text{AMPA}}}{\tau_{\text{AMPA}}} + \sum_j w^{ij}_{\text{AMPA}} \delta(t - t_j),$$

$$\dot{g}^i_{\text{NMDA}} = -\frac{g_{\text{NMDA}}}{\tau_{\text{NMDA}}} + \sum_j w^{ij}_{\text{NMDA}} \delta(t - t_j),$$

$$\dot{g}^i_{\text{GABA}_A} = -\frac{g_{\text{GABA}_A}}{\tau_{\text{GABA}_A}} + \sum_j w^{ij}_{\text{GABA}_A} \delta(t - t_j). \qquad (8)$$

**Appendix table 1**. Parameter values for synapses

| Name | Units | Value |
|---|---|---|
| $E_{\text{AMPA}}$ | mV | 0 |
| $\tau_{\text{AMPA}}$ | ms | 1 |
| $E_{\text{NMDA}}$ | mV | 0 |
| $\tau_{\text{NMDA}}$ | ms | 100 |
| $E_{\text{GABA}_A}$ | mV | −75 |
| $\tau_{\text{GABA}_A}$ | ms | 5 |

After each spike of a presynaptic neuron $j$, each corresponding conductance was incremented by $w^{ij}$.

In MEC layer II, basket cells receive a potent, NMDA-mediated synaptic excitation (**Jones and Buhl, 1993**). These NMDA responses are slow, lasting several tens of ms (**Jones and Buhl, 1993**). NMDA synapses in the attractor network are thus represented by an exponentially decaying conductance ($g_{\text{NMDA}}$), with a 100 ms time constant (**Appendix table 1**). Both the voltage dependence and slow kinetics of NMDA receptors have been suggested to help maintain persistent activity in working memory networks (**Wang, 1999**). Here, it is the slow kinetics of $g_{\text{NMDA}}$ that is necessary to maintain the state of the network during consecutive theta cycles. NMDA receptors are known to be of several variants, depending on the types of the subunits the receptors are composed of (**Paoletti et al., 2013**). These several receptor variants have different kinetic time scales, and different sensitivity to the concentration of Mg$^{2+}$. In **Jones and Buhl (1993)**, the authors do not report, quantitatively, to what extent the amplitude of the NMDA-mediated synaptic responses are dependent on the Mg$^{2+}$ concentration. Therefore, we assume here that the slow kinetics of $g_{\text{NMDA}}$ is sufficient to stabilise the activity of the network and do not include voltage-dependence of NMDA conductances.

Finally, the current external to the neuron

$$I_{\text{ext}}(t) = I_{\text{const}}(t) + I_{\theta}(t) + I_{\text{vel}}(t) + I_{\text{place}}(t), \qquad (9)$$

consists of a constant value ($I_{const}$), a theta modulated part, modeled as

$$I_\theta(t) = \frac{A_\theta}{2}(1 + \sin(2\pi f_\theta t + \phi_\theta)), \qquad (10)$$

the velocity modulated current ($I_{vel}$) that simulates a combination of head-direction input and animal speed input, and an input coming from place cells ($I_{place}$). The description of the parameters in the equations can be found in **Appendix table 2**. The theta current drive is the sum of a constant amplitude positive current ($I_{const}$) and a current with sinusoidal waveform ($I_\theta$). The constant component of the drive is required to activate the circuit. If it is removed then the circuit becomes silent. The sinusoidal waveform has a frequency of 8 Hz. This is chosen to reflect the frequency of theta oscillations in behaving animals. The amplitude is chosen to produce theta modulation of interneuron firing similar to that observed in behaving animals (cf. **Chrobak and Buzsáki, 1998**) and in ex-vivo models of theta-nested gamma activity (cf. **Pastoll et al., 2013**). While $I_{const}$ and $I_\theta$ are simple functions of time, the velocity modulated current and place cell current are described separately. The velocity modulated current is described in 'Velocity modulated input current' and the place cell input current in 'Place cell input'.

**Appendix table 2**. Neuron parameters and their description

| Name | Description | Name | Description |
|---|---|---|---|
| $V_m$ | Membrane potential | $E_{AMPA}$ | AMPA reversal potential |
| $C_m$ | Membrane capacitance | $g_{NMDA}$ | NMDA conductance |
| $g_L$ | Leak conductance | $E_{NMDA}$ | NMDA reversal potential |
| $E_L$ | Leak reversal potential | $I_m$ | Trans-membrane current |
| $g_{AHP}$ | AHP conductance | $I_{syn}$ | Synaptic current |
| $\tau_{AHP}$ | AHP time constant | $I_{syn}$ | Synaptic current |
| $E_{AHP}$ | AHP reversal potential | $I_{ext}$ | External current |
| $\Delta_T$ | Spike initiation width | $I_{const}$ | Constant current |
| $V_T$ | Spike initiation threshold | $I_\theta$ | Theta-modulated current |
| $g_{GABA_A}$ | GABA conductance | $I_{vel}$ | Velocity current |
| $E_{GABA_A}$ | GABA reversal potential | $I_{place}$ | Place cell current |
| $g_{AMPA}$ | AMPA conductance | $\tau_{AMPA}$ | AMPA time constant |
| $\tau_{GABA_A}$ | GABA time constant | $\tau_{NMDA}$ | NMDA time constant |
| $g_{ad}$ | Adaptation conductance | $\tau_{ad}$ | Adaptation time constant |
| $g_{AHP_{max}}$ | AHP maximal value | $g_{ad_{inc}}$ | Adaptation conductance increase |
| $A_\theta$ | $\theta$-current amplitude | $f_\theta$ | $\theta$-current frequency |
| $\phi_\theta$ | $\theta$-current phase | – | – |
| $w_{AMPA}$ | AMPA synaptic weight | $w_{NMDA}$ | NMDA synaptic weight |
| $w_{GABA_A}$ | GABA synaptic weight | – | – |

For the exact values used in the simulations, refer to **Appendix tables 1**, **3–5**.

**Appendix table 3**. Single neuron parameter values for all cells

| Name | Units | Value (E cells) | Value (I cells) |
|---|---|---|---|
| $C_m$ | pF | 211.389 | 227.3 |
| $E_L$ | mV | −68.5 | −60 |
| $V_T$ | mV | −50 | −45 |
| $V_r$ | mV | −68.5 | −60 |
| $g_L$ | nS | 22.73 | 22.73 |
| $\Delta_T$ | mV | 0.4 | 0.4 |

*Appendix table 3. Continued on next page*

*Appendix table 3. Continued*

| Name | Units | Value (E cells) | Value (I cells) |
|---|---|---|---|
| $E_{AHP}$ | mV | −80 | × |
| $\tau_{AHP}$ | ms | 20 | × |
| $g_{AHP_{max}}$ | nS | 5 | × |
| $\tau_{ad}$ | ms | × | 7.5 |
| $g_{ad_{inc}}$ | nS | × | 22.73 |

**Appendix table 4**. Parameter values for external inputs

| Name | Units | Value (E cells) | Value (I cells) |
|---|---|---|---|
| $I_{const}$ | pA | 300 | 200 |
| $A_\theta$ | pA | 375 | 25 |
| $\phi_\theta$ | rad | $-\pi/2$ | $-\pi/2$ |
| $f_\theta$ | Hz | 8 | 8 |

**Appendix table 5**. Parameter values for synaptic profiles

| Name | Units | Value |
|---|---|---|
| $\mu$ | normalised | 0.433 |
| $\sigma_{exc}$ | normalised | 0.0834 |
| $\sigma_{inh}$ | normalised | 0.0834 |
| C | normalised | 0.03 |
| $\lambda_{grid}$ | cm | 60 |

## Synaptic connection profiles

In the majority of the simulations the attractor model simulates only connections from E to I cells and vice versa. Synapse strengths of connections originating from E cells are generated by a Gaussian-like function with values dependent on the distance between a presynaptic ($j$) and postsynaptic ($i$) cell on the twisted torus:

$$w_{AMPA}^{ij} = g_E \exp\left(\frac{-\left(d\left(i,j,C,\mathbf{e}_P^j\right) - \mu\right)^2}{2\sigma_{exc}^2}\right), \tag{11}$$

$$d\left(i,j,C,\mathbf{e}_P\right) = \left|\mathbf{u}_i - \mathbf{u}_j - C\mathbf{e}_P\right|_{torus}, \tag{12}$$

$$w_{NMDA}^{ij} = C_{NMDA}\ w_{AMPA}^{ij}. \tag{13}$$

In these equations, $\mu$ is the distance of the excitatory surround from the position of presynaptic neuron, $\sigma_{exc}$ is the width of the excitatory surround, $\left|\cdot\right|_{torus}$ is a distance on the twisted torus that takes the boundaries of the torus into account and $C$ is the synaptic profile shift. The excitatory connections are composed of the equivalent amount of NMDA synaptic conductances. The synaptic strengths of NMDA is specified by a fractional constant $C_{NMDA}$. In all simulations, the NMDA conductance constituted 2% of the AMPA conductance, which was an amount necessary to retain the information about the position of the bump attractor during consecutive theta cycles, while not too high to prevent generation of nested gamma oscillations. In **Equation 12**, $\mathbf{e}_P$ determines the shift of the center of the outgoing synaptic strength profile on the torus, and was used to couple the velocity of the bump with the animal velocity (**Burak and Fiete, 2009**; **Pastoll et al., 2013**). The velocity modulated input is described in more detail in 'Velocity modulated input current'.

Synapse strengths from I cells to E cells in networks with structured connections were generated by a Gaussian function

$$w^{ij}_{\text{GABA}_\text{A}} = g_\text{I}\exp\left(\frac{-d(i,j,0,0)^2}{2\sigma^2_\text{inh}}\right), \tag{14}$$

that takes a distance between the pre- and post-synaptic neurons ($d(i, j, 0, 0)$) and a width of the Gaussian ($\sigma_\text{inh}$) as parameters. As can be seen from **Equation 14**, inhibitory neurons do not have shifts in their outgoing synaptic profiles. In addition, a distance-independent I → E inhibitory connectivity was generated for which the probability of connection between the pre- and post-synaptic cell was 0.4 and the weight of a connection was set to $0.013g_\text{I}$. The total inhibitory synaptic weight was thus a sum of $w_{\text{GABA}_\text{A}}$ in **Equation 14** and the distance-independent component. In simulations with recurrent I → I connectivity (E-I-I networks), I neurons were mutually connected with a connection probability of 0.1 and a constant synaptic weight of 69 pS.

In networks that contain recurrent E → E connectivity, the connections between E cells were modelled as a Gaussian function, that is, similarly to **Equation 14**:

$$w^{ij}_{\text{E} \rightarrow \text{E}} = g_{\text{E}\rightarrow\text{E}}\exp\left(\frac{-d\left(i,j,C,\mathbf{e}^j_p\right)^2}{2\sigma_{\text{E}\rightarrow\text{E}}{}^2}\right), \tag{15}$$

where $C$, $\mathbf{e}_p$ and $\sigma_{\text{E} \rightarrow \text{E}}$ have the same meaning as in **Equation 11**. In these simulations, if not stated otherwise, $g_{\text{E} \rightarrow \text{E}} = 0.5$ nS.

We also evaluated networks in which E → I and I → E synapses were unstructured and have a constant value. Here, the E → E synaptic weights were set according to **Equation 15** and the excitatory and inhibitory synaptic weights for E → I and I → E synapses were set to $g_\text{E}/d$ and $g_\text{I}/d$ respectively, where $d$ is a probability of connection between the presynaptic and postsynaptic neuron, set to 0.1. The density factor $d$ was used in order to ensure equivalence of total synaptic input of a postsynaptic cell when compared to networks that have all-to-all connectivity (**Equations 11, 14, 15**).

Finally, in networks where connection strengths were generated probabilistically instead of in an all-to-all way, the synaptic weights from E to I cells and vice versa were all constant and set to $g_\text{E}$ and $g_\text{I}$ respectively, while the probability of connection between the pre- and post-synaptic neuron was drawn according to **Equation 11** for E → I synapses, and **Equation 14** for I → E synapses.

## Velocity modulated input current

All simulations of grid fields and estimations of the velocity input gain contain current input modulated by the speed and direction of the simulated animal. Although translational activity can be achieved by inputs to either of the populations (**Pastoll et al., 2013**), here we have simulated velocity modulated inputs only onto the E cell population. All E cells are assigned a preferred direction vector (**Equation 12**) that shifts the outgoing synaptic profile in the direction specified by the unit vector $\mathbf{e}_p$ in **Equation 12**. The preferred directions are drawn from a set of four unit vectors pointing up, down, left and right so that all directions are distributed along the twisted torus.

During simulated movement of the animal, the velocity modulated current injected into the neuron $i$ is computed as follows (here · is a dot product):

$$I^i_{vel}(t) = C_v\mathbf{v}(t) \cdot \mathbf{e}^i_p,$$

$$C_v = \frac{N_x}{a\lambda_\text{grid}}. \tag{16}$$

The gain of the velocity input ($C_v$) is determined from the number of neurons the bump needs to translate in order to return to the original position ($N_x$ [neurons]; on a twisted torus this quantity is effectively the horizontal size of the neural sheet) divided by the product of the expected grid field spacing ($\lambda_{grid}$ [cm]) and a slope of the relationship between bump speed and injected velocity current magnitude ($a$ [neurons/s/pA]). Therefore, given a desired spacing between grid fields, the gain of the velocity inputs can be calibrated.

## Place cell input

Because of the finite network size, spiking variability, or imperfections in the synaptic profile functions, the position of bump attractor in the network might drift over time. The simulations of grid firing fields (*Figures 2, 6D–I, 7A–C* and associated figure supplements) and simulations that explored the controllability of the network by place cell input (*Figure 6—figure supplement 5*) included a separate population of cells with place-like firing fields connected to E cells (in all other simulations the input was de-activated). Inputs from these cells opposed drift of the bump attractor.

Place cells were simulated as independent inhomogeneous Poisson processes, whose rate was modulated by a Gaussian function of the simulated animal location. Thus, the firing rate of an $i$th place cell, $r_i$ was:

$$r_i = r_{max} \exp\left(-\frac{|\mathbf{l} - \boldsymbol{\mu}_i|^2}{2\sigma_{field}^2}\right),$$

(17)

where $r_{max}$ is firing rate in the center of the place field, $\mathbf{l}$ is an instantaneous position of the simulated animal, $\boldsymbol{\mu}_i$ is the center of the place field and $\sigma_{field}$ is the width of the place field. In all simulations, there were 900 place cells, with $r_{max} = 50$ Hz, and $\sigma_{field} = 20$ cm. Spikes emitted by place cells were thus generated by independent Poisson processes with rate $r_i(t)$ in *Equation 17*, and the centres of individual place fields were uniformly distributed in the arena the simulated animal was moving in. The connection weights from place cells were arranged in a divergent manner, so that a place cell had strongest connections with grid cells whose firing fields were aligned (in real space) with the firing field of the place cell. The connection weight from place cell $i$ to a grid cell $j$ decayed according to a Gaussian function

$$g_{ji} = G_{PC}^{max} \exp\left(-\frac{\left|\boldsymbol{\mu}_{PC}^i - \boldsymbol{\mu}_G^j\right|^2}{2\sigma_{PC}^2}\right),$$

(18)

where $G_{PC}^{max}$ is the maximal connection strength between two fully aligned grid and place fields, $\boldsymbol{\mu}_{PC}^i$ is the centre of the place field of the $i$th place cell, $\boldsymbol{\mu}_G^j$ is the centre of the grid field of the $j$th grid cell that is nearest to the place cell, $\sigma_{PC}$ is the width of the synaptic profile. The parameters were set to $G_{PC}^{max} = 0.5$ nS and $\sigma_{PC} = 7$ cm. Connections from place cells were modelled as AMPA conductances only (*Equation 8*). This was sufficient for the purpose of opposing drift of the bump attractor and we therefore do not include any more biological detail into these connections.

In simulations where I cells received uncorrelated spatial inputs (*Figure 2—figure supplement 4*), an additional population of place cells was instantiated, with parameters set to $r_{max} = 100$ Hz and $\sigma_{field} = 80$ cm. Each I cell received connections from three randomly chosen place cells, with a synaptic weight of 4 nS.

## Bump attractor initialisation

Each simulation contains an initialisation stage that attempts to set the model into the desired state, that is, a bump attractor. During this stage, the theta-modulated input is switched off and the network receives only the constant input source (see *Equation 9*). The bump attractor might not form spontaneously, and instead the network could persist in a uniform firing rate regime (*Compte et al., 2000*). However, it might be possible that when forced into the attractor state, the network will persist (data not shown). Therefore, we used the place cell input as a spatially-tuned

input that served (i) as an initialisation input in order to drive the network into an attractor state if this does not happen spontaneously and (ii) to initialise the bump attractor position so that the phase of grid firing fields matched the positions of place fields. The initialisation phase lasted for the first 500 ms of simulation time, during which the firing rate of place cells were doubled, and the strength of connections from place cells to grid cells was increased 10-fold.

## Parameter space exploration

The excitatory and inhibitory parameter space exploration was performed by varying the amount of inhibitory and excitatory synaptic strengths. Since the actual synaptic weights are a function of distance on the twisted torus, we used the maximal conductance of AMPA ($g_E$, **Equation 11**) and GABA synapses ($g_I$, **Equation 14**) in all the parameter exploration plots. Note that since the amplitude of NMDA conductances was a fixed fraction of that of AMPA, the strength of NMDA was also scaled as a function of $g_E$ in line with the scaling of the AMPA conductance and was thus implicitly counted toward $g_E$. Additionally, in **Figure 7—figure supplement 10**, parameter exploration simulations were performed in which the $g_{E \to E}$ synaptic scaling variable, as well as the width of the synaptic profile of E → E connections ($\sigma_{E \to E}$ in **Equation 15**) was used.

## Analysis of spatial firing fields

Gridness scores were calculated following previous studies (**Sargolini et al., 2006**), by taking the spatial autocorrelation of each firing field (a region corresponding to a circle with radius $\lambda_{grid}/2$ and a centre in the middle of the autocorrelation function has been removed) and rotating in steps of three degrees. For each rotation a Pearson correlation coefficient was calculated with the original autocorrelation. To calculate the gridness score the maximum of values at 30, 90 and 150° rotation was subtracted from the minimum of the values at 60 and 120° rotation.

Spatial information (bits/spike) was calculated according to (**Skaggs et al., 1996**):

$$I = \sum_{i=1}^{N} p_i \frac{\lambda_i}{\lambda} \log_2 \frac{\lambda_i}{\lambda}, \tag{19}$$

where the environment was divided into $N$ bins and $p_i$ was the occupancy probability of bin $i$, $\lambda_i$ was the mean firing rate for bin $i$ and $\lambda$ was the overall mean firing rate of the cell.

Spatial sparsity was calculated following (**Buetfering et al., 2014**):

$$S = 1 - \frac{\left( \sum_{i=1}^{N} p_i \lambda_i \right)^2}{\sum_{i=1}^{N} p_i \lambda_i^2}, \tag{20}$$

where $N$, $p_i$ and $\lambda_i$ have the same meaning as in **Equation 19**.

## Estimating gain of the velocity-dependent inputs

In order to estimate the precision of velocity integration in a continuous attractor, we have performed shorter simulations in which a constant velocity input (in a vertical direction) was injected into E cells for a period of 10 s. Based on this set of simulations, the slope of the relationship between bump speed and the injected velocity current was estimated (in units of neurons/s/pA). The estimation was based on the following algorithm:
1. Estimate the range of bump speeds that need to be covered (**Appendix figure 1**).

$$s_{bump}^i = \frac{N_x}{\lambda_{grid}} s_{animal}^i, \tag{21}$$

where $s_i$ are the speeds of the animal/bump, estimated from forward differences of the trajectory of the simulated animal, $N_x$ is the horizontal size of the neural sheet (neurons), and

$\lambda_{grid}$ is the grid field spacing (cm). These speeds will form a distribution of bump speeds that the attractor must achieve in order to path integrate without error (**Appendix figure 1B**).

2. Pick a specified percentile from this distribution (here the 99th percentile was used), that is, the maximal speed of the bump, in order to account for the specified fraction of animal velocities, set this as $s_{max}$. The range of target bump speeds will be $<0, s_{max}>$.

3. For each $I_{vel} \in \{0, 10,…, 100\}$ pA, estimate the bump speed by tracking its position on the neural sheet, using the 'Gaussian fitting procedure'. Repeat this step 10 times. This step acquires data for estimating the relationship between the slope of bump speed and injected velocity current.

4. For each $I_{vel}^{max} \in \{10, 20, …, 100\}$ pA, estimate a line fit on data samples with the velocity current in the range of $I_{vel} \in <0, I_{vel}^{max}>$, that is, fit the line to only a subset of velocity current data points.

5. Remove all fits that do not fit at least $<0, s_{max}>$ on the bump velocity axis.

6. If there are any lines left, select line with the minimal error of fit (normalized by the number of data points used); otherwise select line (from the original list) that covers the maximal range of bump speeds.

7. Calculate the slope of the selected line and finish.

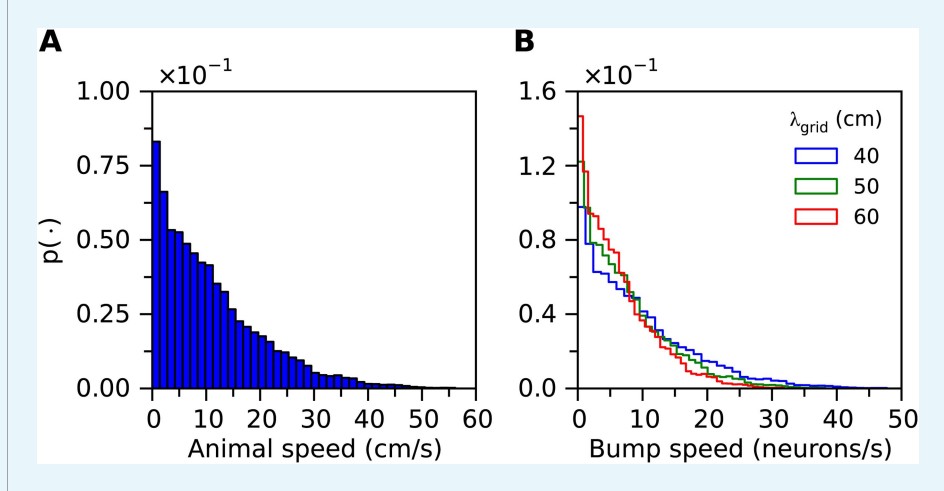

**Appendix figure 1**. (**A**) Histogram of velocities of a simulated animal. (**B**) Histogram of bump speeds derived from the animal velocities estimated in **Equation 21**, for different grid field spacings.

## Simulation protocols

### Simulations of animal movement

Simulations of animal movement were carried out for 600 s of simulated time. Here, for each value of $g_E$ and $g_I$, the main simulation run was preceded by a number of shorter simulations which determined how much current needs to be injected in order for the bump of activity to track the simulated movement of an animal ('Estimating gain of the velocity-dependent inputs'). This procedure calibrates the gain of the velocity input current in order to produce grid fields with a specified spacing between the peaks in the individual firing fields. The result is a single number in units of neurons/s/pA, which determines the speed of the bump as a function of injected velocity input. The spacing between the individual fields of the grid firing fields was set to 60 cm in all of the simulations.

During the main simulation run, the animal movement was simulated for 600 s. Each of the runs was repeated four times for simulations in **Figure 2** and three times for simulations in **Figure 6D–I** and once for networks that contain additional recurrent synapses between E cells or I cells, as well as in networks with synapses generated probabilistically. These simulations use the estimated velocity response gain in order to calibrate the spacing between the grid firing

fields. After the simulation was finished, a neuron in the bottom-left corner of the torus was selected for analysis. For this cell the gridness score of its firing field was computed. The reasoning behind choosing only a single cell to estimate the gridness score is as follows. The grid firing fields in the network are a result of coordination of activity of the network as a whole. If the network forms a bump attractor that is able to accurately track animal movement, all cells in the network will have grid-like firing fields that differ only in their phases. On the other hand, if the bump attractor does not form, is unstable, or does not accurately track the position of the animal, the gridness score of all cells will be low. Thus, the firing field of a single cell in the network represents grid field computation in the network as a whole. Moreover, this cell can be selected arbitrarily. This condition might not hold in simulations where I cells receive uncorrelated spatial inputs and therefore in these simulations firing fields of 100 randomly selected cells from both E and I populations were used to calculate the gridness scores (*Figure 2—figure supplement 4*).

## Short simulation runs without animal movement

Some of the simulation runs were used to estimate properties of bump attractors and nested gamma oscillations. In these experiments, instead of simulating animal movement, a shorter, 10 s simulation, was run. The velocity and place cell input were deactivated. Thus, the network is expected to only produce a static bump attractor and does not perform path integration. For each parameter setting (determined by $g_E$, $g_I$, and the noise level), five simulations were performed. For each simulation run, post-synaptic currents were recorded from 25 randomly selected excitatory cells in the model by clamping their membrane potential at −50 mV (this was done by simulating a separate process for each of the selected neurons, while simulating the original membrane potential according to *Equation 1*). Thus, on each run, different cells could be picked up for analysis. It is in principle possible to record membrane currents from all the neurons. However, the amount of data generated by such simulations quickly becomes overwhelming (on the order of several terabytes per parameter exploration experiment). Thus the approach chosen here was to sample from the population of neurons and store the recorded state variables of only a subset of these. This allowed for unrestricted analysis and visualisation of the recorded state variables.

## Analysis of nested gamma oscillations

We estimated the properties of nested gamma oscillations by using autocorrelation functions of the inhibitory currents impinging from inhibitory synapses onto excitatory cells. These currents were estimated from 25 randomly selected excitatory cells recorded during the simulation run. For each neuron, the current was then band-pass filtered between 20 and 200 Hz, the autocorrelation function was computed and then used to detect local maxima after excluding the first peak. The positions of local maxima were calculated as those points in the autocorrelation function where the first difference of the signal changed sign from positive to negative and thus approximated points where the first derivative is zero and the second derivative is negative. The power and frequency of the underlying oscillation was then estimated from the correlation value and from the time lag of the first detected autocorrelation peak respectively. Both values were averaged over all 25 recorded neurons and then subsequently averaged over all simulation trials.

## Gaussian fitting procedure

In networks where properties of bump attractors, such as the position and presence of an activity bump, were estimated, we developed a procedure to fit Gaussian functions onto successive snapshots of network activity of E cells. The network activity snapshots were estimated by taking action potential times of all E cells and estimating their immediate firing rate using a 250 ms wide sliding window with a 125 ms time step. For each snapshot, the properties of a bump-like network activity (if it was a bump) were then estimated by fitting a symmetric Gaussian function to the network activity snapshots, using the maximum likelihood estimator under Gaussian noise (the least squares fitting method):

$$B(\mathbf{X}) = A\exp\left(-\frac{\left\|\mathbf{X}-\boldsymbol{\mu}\right\|^2}{2\sigma_{\text{bump}}^2}\right), \tag{22}$$

where $A$ was the height of the Gaussian function, $\mathbf{X}$ was neuron position on the twisted torus, $\boldsymbol{\mu}$ was the centre of the Gaussian, $\sigma_{\text{bump}}$ was the width of the Gaussian, and $\|\cdot\|$ represents a distance metric on the twisted torus. The parameters fitted were $A$, $\boldsymbol{\mu}$ and $\sigma_{\text{bump}}$. These parameters were then used as the basis for further analysis.

