## [Decision Letter]

[Editors’ note: this article was originally rejected after discussions between the reviewers, but the authors were invited to resubmit after an appeal against the decision.]

Thank you for choosing to send your work entitled “Noise promotes independent control of gamma oscillations and grid firing within a recurrent attractor network” for consideration at *eLife*. Your full submission has been evaluated by Eve Marder (Senior editor) and three peer reviewers, one of whom is a member of our Board of Reviewing Editors, and the decision was reached after discussions between the reviewers. Based on our discussions and the individual reviews below, we regret to inform you that your work will not be considered further for publication in *eLife* at this time.

While all the reviewers felt that the work was intriguing and could be of interest to the gamma oscillation, noise and grid cell community, there was concern that the present model was not consistent with existing experimental data. Specifically, discrepancies regarding inhibitory cell connectivity and recent experimental work (4) need to be addressed. It was felt that this would require re-doing and/or performing additional simulations to determine whether results remained the same. Further detailed comments for consideration are provided below. While you may choose to resubmit a revised manuscript, please note that substantial progress to address the reviewers' concerns is required for a subsequent review to be considered.

*Reviewer #1*:

The authors study the noise-sensitivity of a model previously introduced in a Neuron paper. The connectivity is different from other models for grid cells because there is no recurrent excitation; rather the stellate cells interact via inhibitory cells. To obtain nice oscillations it is helpful that cells have similar firing rates, however to encode information in the level of activity it is necessary that cells have different firing rates. The authors determine under what conditions both can be achieved simultaneously and find that there is an optimal noise level. The mechanism described bears some resemblance to study of an interneuron network by [43] wherein noise reduces the effects of heterogeneity in firing rate on the level of gamma synchronization and increases the range of conductances for which oscillations are obtained, but at the price of neurons not firing on each cycle, similar to the results shown in the manuscript. The results are nice and I found the paper interesting.

In Figures 2 and 4 place fields are shown. I presume they are from E cells. What do the place fields for I cells look like? Does this conform to experimental data?

From the Neuron paper I gather that there is no recurrent excitation, but what is the evidence for the absence of mutual inhibition?

The neurons are connected all to all. Would an actual stochastic connectivity according to a probability that is scaled to the conductance pattern shown in Figure 1, also work and provide the appropriate noise level?

The authors focus on varying synaptic strength g_I_ and g_E_, which shows the robustness of oscillations/attractor states, but I would expect that synaptic strength of the network in vivo would not vary that much over time, or could that happen through synaptic plasticity effects? It would be nice to discuss the relevance of the conductance range investigated.

*Reviewer #2*:

I have only one major comment. This work is largely based on this lab's prior finding that layer II stellate cells do not show recurrent connectivity. This was a problem for standard attractor model connectivity, so the authors conceived the E-I-E attractor model. I think this is great. But, it remains possible that layer II pyramidal cells do show recurrent connectivity, which would support the standard attractor models. There is now controversy which cell types correspond to grid cells, and thus far the published (recent Neuron paper from Brecht's lab) and unpublished (David Roland's poster at SfN from the Moser lab) data suggest that both cell types in layer II are grid cells. Since pyramidal cells might/probably show recurrent connectivity, I would like to know if the same principles about noise emphasized by this paper would also apply to more standard E–E attractor models. I wouldn't ask the authors to explicitly model this, but do the authors have any insight about this that they could add to the Discussion? This is especially relevant to their very broad discussion about the beneficial role of noise in neural computation in general and the protective role of noise against seizures. Most cortical circuits have recurrent excitatory connectivity. Perhaps the authors could address more standard E–E attracter that contains global inhibition in their discussion.

*Reviewer #3*:

The authors created a recurrent attractor network, and investigated how noise affected synaptic activity at gamma frequencies and grid firing. Their recurrent attractor network incorporates E-I-E connectivity to produce grid firing through a velocity-dependent update of network attractor states, and also produce theta-nested gamma frequencies. They found that noise can increase the range of synaptic strengths with which gamma activity and grid computations are produced, and that synaptic gamma frequency and amplitude can be modulated independently from the grid firing.

In general, the manuscript was well-written, and the results logically follow from their model. However, fundamental questions about the model itself are important to consider. Recently, Buetfering and colleagues found that parvalbumin-positive (PV+) interneurons in the medial entorhinal cortex (MEC) integrate input form grid cells with various phases, and exhibited low spatial sparsity and no spatial periodicity ([4]. Nat Neurosci 17(5):710–718.). This argues against a scenario in which fast-spiking interneurons mediate grid cell phase-dependent recurrent inhibition in the MEC – a necessary component of the authors' recurrent attractor network model, where the activity bump in the E-cell population is reflected by an inverted bump in I-cell population activity. This discrepancy needs to be addressed.

[Editors’ note: what now follows is the decision letter after the authors submitted for further consideration.]

Thank you for resubmitting your work entitled “Noise promotes independent control of gamma oscillations and grid firing within recurrent attractor networks” for further consideration at *eLife*. Your revised article has been favorably evaluated by Eve Marder (Senior editor), a Reviewing editor, and two additional reviewers.

The manuscript has been significantly improved and only minor revisions are required. Once you have addressed these trailing issues, the manuscript should be acceptable without further review.

*Reviewer #1 Minor Comments*:

I have one minor question, in their new Figure 2, they show unimodal spatial autocorrelation images of the I cells. This does occur in the Buetfering paper Figure 4, but there are also a few with a weak periodic pattern (lower right panel of Figure 4, grid score 0.18). Can this be found in the simulations as well?

*Reviewer #2 (General assessment and major comments (Required))*:

The authors responded to my comments adequately. In addition to the initial goal of exploring conductance parameters from E-I and I-E, the manuscript now provides substantially additional simulations to support E-I-E attractor models in general.

In their rebuttal the authors defend against the criticism that experimental data has not shown grid firing fields in inhibitory neurons. This criticism manifests from Buettfering et al., 2014 who report that PV interneurons do not show grid firing patterns. The authors have completed additional analyses and expanded their discussion to argue that these experimental data are no inconsistent with E-I models. (1) I cells in their model are weakly tuned grid cells (often below the gridness threshold used by Buettfering). (2) New simulations show that I cells show less grid-like activity when they are connected to uncorrelated grid cells. (3) The authors point out that [4] only examine PV interneurons, and thus, other interneurons could exhibit grid firing fields. Overall I find these arguments convincing (especially #2), which fuel the debate about the structure of plausible E-I models and lead to new predictions (non-spatially correlated input to I cells, and/or other interneuron classes show grid properties). Close examination of experimental data show clear grids and not-so-clear grids (Krupic, Burgess, and O'Keefe, 2012) which may to correspond to firing on different theta phases (Newman and Hasselmo, 2014), similar to many interneuron classes in CA1 of the hippocampus.

The authors add I–I connectivity, consistent with their experimental data, and the model is robust. This suggestion and implementation has strengthened the manuscript.

My concern about E–E models has been addressed. While I do believe E–E models are still possible the authors correctly argue that this issue is far from settled. We do not yet have a clear understanding of MEC microcircuits, especially intralaminar connections that may or may not be needed for grid cell generation. However, the authors add additional simulations of E-E-I models, which provide similar results to the E-I models.

*Reviewer #3*:

Overall, we think that the authors have addressed our concerns. In particular, the additional simulations and discussions about how their findings relate to [4], examining the effect of recurrent inhibition (I–I) has made the paper much stronger, and tying their results into some experimentally testable predictions.

The general results showing how different mechanisms could control grid activity and gamma oscillations are interesting, although a bit more of an explanation of the precise mechanisms of their gamma oscillation is needed (i.e., first paragraph of Discussion).

---

## [Author Response]

[Editors’ note: the author responses to the first round of peer review follow.]

We appreciate the initial enthusiasm of the reviewers for the manuscript and recognise their concerns regarding possible discrepancies between inhibitory cell connectivity and recent experimental work. In recognition of the editorial view that addressing these concerns would require “re-doing and/or performing additional simulations to determine whether results remained the same”, we have carried out substantial additional simulations and analysis that we believe addresses in full the issues raised by the reviewers and further strengthens the conclusions of our study. Our additional work includes the following new advances:

1) To address the concern that there may be discrepancies between our proposed models for grid firing and recent data from Buetfering et al. we have adopted two approaches. First, we analysed in detail the spatial firing properties of interneurons in networks that we simulate. We find that these properties are in fact consistent with the Buetfering et al. data set. Importantly, our new results provide substantial benchmarks against which future experimental data can be evaluated. Second, whereas previously we did not directly address the results of Buetfering et al., in a revised manuscript we will present our new analysis and will also highlight that the Buetfering study focuses only on a subset of inhibitory neurons that are labelled by parvalbumin. It remains to be determined whether these, or another population of interneurons, is more relevant to the predictions made by the models that we propose.

2) To address the concern about the extent to which our results hold if known connections between interneurons are incorporated into our model, we have carried out additional simulations with models that incorporate this connectivity. While our initial model was deliberately simplified to facilitate its analysis, we now show that our main conclusions nevertheless hold when inhibitory connections between interneurons are accounted for. These new simulations also demonstrate for the first time that this additional inhibitory connectivity increases the robustness of grid firing and the frequency of gamma oscillations.

3) To address concerns about scenarios in which connections between excitatory neurons may play important roles in grid firing, for example if pyramidal rather than stellate cells are the primary generators of grid activity, we have adopted two approaches. First, we have implemented new models in which structured coupling between excitatory cells is incorporated along with the existing structured interactions between excitatory and inhibitory cells. Our new data demonstrate that this scenario has little impact on our main conclusions. Second, we evaluate scenarios in which only connections between excitatory neurons are structured. We find that in this scenario excitatory feedback causes circuits to become unstable and as a result we are unable to identify network configurations that support grid firing. This is consistent with previous theoretical work demonstrating instability of much simpler attractor networks based exclusively on structured excitation. As these new data suggest that structured connections between excitatory neurons are on their own unlikely to support grid firing in networks of spiking neurons, they further support our proposal that grid attractors are primarily generated through interactions between excitatory and inhibitory cells.

4) We have carried out new simulations that address concerns about the detailed implementation of our models. For example, we have evaluated models in which connections are probabilistic. Results of simulations with these models are in line with our original conclusions.

Given that our new data and analysis address the reviewers' previous concerns in full, and provide further strong support for our initial conclusions, we would like to submit to *eLife* a revised version of our manuscript incorporating these findings.

We continue to believe that our results will be of interest to a broad and diverse audience. Our study offers novel and general insights into the relationship between gamma oscillations and neural computation, and their dependence on underlying synaptic mechanisms. By establishing direct causal relationships between synaptic mechanisms, computation and gamma oscillations our results will guide future investigation of normal cognitive states, and of disorders including autism and schizophrenia. We hope that given the substantial new work that we have carried out it will be suitable for publication in *eLife.*

Reviewer #1:

*The authors study the noise-sensitivity of a model previously introduced in a Neuron paper. The connectivity is different from other models for grid cells because there is no recurrent excitation; rather the stellate cells interact* via *inhibitory cells. To obtain nice oscillations it is helpful that cells have similar firing rates, however to encode information in the level of activity it is necessary that cells have different firing rates. The authors determine under what conditions both* can *be achieved simultaneously and find that there is an optimal noise level. The mechanism described bears some resemblance to study of an interneuron network by*
[43]
*wherein noise reduces the effects of heterogeneity in firing rate on the level of gamma synchronization and increases the range of conductances for which oscillations are obtained, but at the price of neurons not firing on each cycle, similar to the results shown in the manuscript. The results are nice and I found the paper interesting*.

We apologize for the oversight and now cite the study by [43] in the Discussion. In their study of stochastic weak synchronization in interneuron networks, noise is required for emergence of oscillations, but increasing noise suppresses oscillations. Nevertheless, the model considered by [43] differs in important ways from the models we investigate here. In particular, it does not generate network attractor states or appear to carry out a rate-coded computation, and it does not include excitatory neurons.

*In*
Figures 2 and 4
*place fields are shown. I presume they are from E cells. What do the place fields for I cells look like? Does this conform to experimental data*?

Figure 2 shows spatial firing fields from E cells and Figure 4 shows network activity of E cells. We have modified the figure legends to make this clear**.**

We did not previously show the I cell fields because our focus was on grid firing and we previously reported predicted spatial firing properties of I cells in E-I networks (e.g. Figure 7 in [30]). Given the interest of the reviewer we have now amended Figure 2 to show examples of I cell fields alongside the firing fields of the E cells.

The previous version of our manuscript also did not address the relationship of the firing fields of I cells to experimental data. While not an initial focus of this study, we now include substantial new data, analysis and discussion to address this. Our major changes relevant to this issue are as follows.

1) Only the spatial firing fields of parvalbumin (PV) positive interneurons have been examined experimentally (see Buetfering et al., Nature Neuroscience, 2014). These PV cells have firing fields with significant spatial stability, but compared with grid cells they on average have much lower spatial sparsity and grid scores. These observations were interpreted by Buetfering et al. as evidence against E-I models, which we showed previously to predict spatial firing by I cells (cf. [30]). However, before reaching this conclusion it is important to first consider carefully the predictions of E-I models. We now report additional analysis which demonstrates that in E-I models that generate grid firing, the firing fields of I cells have substantially lower spatial sparsity, spatial information and grid scores than E cells (Figure 2 and Figure 2—figure supplement 2 and Figure 2—figure supplement 3. Importantly, the grid scores of I cells in all network configurations are less than the corresponding E cells and in many network configurations are <0.4, which is below the cut off applied by Buetfering et al. to identify grid firing.

2) To obtain a better understanding of the robustness of spatial firing by I cells to interference from out of field spatial input, and of the distribution of spatial firing within an E-I network, we have also simulated E-I networks in which I cells receive uncorrelated spatial input. We find that grid firing in E-I networks is robust to substantial uncorrelated spatial inputs. Importantly, in these simulations we find further reductions in the rotational symmetry of interneuron firing fields, and as a result a further reduction of their grid scores. We report this new data in Figure 2–figure supplement 6. Together, these results establish that E-I networks can generate grid firing by E cells in conditions in which I cells have spatial firing that does not satisfy current metrics for gridness. Thus, the Buetfering data set is in fact consistent with the properties of I cells in E-I attractor networks.

3) While the Buetfering et al. data clearly show spatial firing of PV-interneurons and as we show above are broadly consistent with our model predictions, the firing properties of other classes of interneuron in the MEC are unknown. Thus, it remains to be determined if other types of interneurons present in layer II can exhibit spatial firing fields predicted by E-I networks as these have not been recorded from in the study by [4]. We now make this issue clear in the Discussion.

In summary, the experimental data published to date is consistent with E-I classes of model that we consider here. Importantly, our new analysis provides substantial benchmarks against which future experimental data can be evaluated.

*From the Neuron paper I gather that there is no recurrent excitation, but what is the evidence for the absence of mutual inhibition*?

Our experimental observations indicate that mutual inhibition is likely to be present between interneurons that also connect with stellate cells (cf. Pastoll et al. Figure 3). As the reviewer notes, this was not included in our original model. This was primarily to reduce the complexity of the model and therefore make analysis more tractable.

To address this we have carried out new simulations in which I–I connectivity is included in the network. In these simulations our primary qualitative conclusions hold (see Figure 7 and Figure 7—figure supplement 1, Figure 7—figure supplement 10, Figure 7—figure supplement 2, Figure 7—figure supplement 3, Figure 7—figure supplement 4, Figure 7—figure supplement 5, Figure 7—figure supplement 6, Figure 7—figure supplement 7, Figure 7—figure supplement 8 and Figure 7—figure supplement 9). Interestingly, in addition we find that addition of I–I connections causes an overall increase in the frequency of gamma oscillations (see Figure 7). This is important because previously the gamma oscillations in our simulated E-I networks had frequency at the lower end of that observed experimentally (cf. Chrobak and Buzsaki, 1998; [30]), whereas with more realistic networks incorporating I–I coupling the gamma frequency matches the typical experimental observations more closely.

*The neurons are connected all to all. Would an actual stochastic connectivity according to a probability that is scaled to the conductance pattern shown in*
Figure 1*, also work and provide the appropriate noise level*?

This is an interesting idea. To address it we have simulated networks with the suggested stochastic connectivity scaled and a fixed synaptic weight for each synapse whose value depends on g_E_ and g_I_ for E and I connections respectively (Figure 1—figure supplement 1). This model reproduces our results obtained previously by simulating models with the all-to-all connectivity. We have now included in the manuscript an illustration of synaptic weights for both simulated cases (Figure 1—figure supplement 1), and the results of the new simulations which demonstrate the role of noise, g_E_ and g_I_ in models with probabilistic connectivity (Figure 2—figure supplement 3 and Figure 2—figure supplement 4, Figure 3—figure supplement 1, Figure 4—figure supplement 1, Figure 5–figure supplement 2, Figure 6–figure supplements 7 and 8).

*The authors focus on varying synaptic strength g*_*I*_
*and g*_*E*_*, which shows the robustness of oscillations/attractor states, but I would expect that synaptic strength of the network* in vivo *would not vary that much over time, or could that happen through synaptic plasticity effects? It would be nice to discuss the relevance of the conductance range investigated*.

We now add to the Discussion that “Adjustment of g_E_ and g_I_ could be achieved dynamically through actions of neuromodulators (26), or on slower developmental time scales (47)”.

Reviewer #2:

*I have only one major comment. This work is largely based on this lab's prior finding that layer II stellate cells do not show recurrent connectivity. This was a problem for standard attractor model connectivity, so the authors conceived the E-I-E attractor model. I think this is great. But, it remains possible that layer II pyramidal cells do show recurrent connectivity, which would support the standard attractor models. There is now controversy which cell types correspond to grid cells, and thus far the published (recent Neuron paper from Brecht's lab) and unpublished (David Roland's poster at SfN from the Moser lab) data suggest that both cell types in layer II are grid cells. Since pyramidal cells might/probably show recurrent connectivity, I would like to know if the same principles about noise emphasized by this paper would also apply to more standard E–E attractor models. I wouldn't ask the authors to explicitly model this, but do the authors have any insight about this that they could add to the Discussion? This is especially relevant to their very broad discussion about the beneficial role of noise in neural computation in general and the protective role of noise against seizures. Most cortical circuits have recurrent excitatory connectivity. Perhaps the authors could address more standard E–E attracter that contains global inhibition in their discussion*.

We agree that based on current data stellate and pyramidal cells could both be grid cells, but this issue is far from settled and their relationships to one another are not clear. It is likely that pyramidal cells make excitatory connections to one another (Dhillon and Jones, 1999; [8]). There is evidence that pyramidal cells also interact via inhibitory interneurons (Varga et al., 2010) and therefore pure E–E models are unlikely to be applicable. However, it is possible models in which E–E coupling co-exists with E- > I and I- > E coupling might also be consistent with experimental data. We have therefore explored this issue with additional simulations. The results of these simulations of E-E-I networks are similar to our findings with E-I networks. We include the results of these simulations and additional discussion in the revised manuscript (Figure 7–figure supplements 6–11).

For completeness we have also carried out simulations to evaluate networks in which only connections between excitatory neurons are structured, while interactions between excitatory and inhibitory neurons are uniformly distributed across the network. We find that the resulting unconstrained excitatory feedback causes these networks to be unstable, preventing generation of bump attractors and gamma activity (Figure 7–figure supplement 12).

Reviewer #3:

*The authors created a recurrent attractor network, and investigated how noise affected synaptic activity at gamma frequencies and grid firing. Their recurrent attractor network incorporates E-I-E connectivity to produce grid firing through a velocity-dependent update of network attractor states, and also produce theta-nested gamma frequencies. They found that noise* can *increase the range of synaptic strengths with which gamma activity and grid computations are produced, and that synaptic gamma frequency and amplitude* can *be modulated independently from the grid firing*.

*In general, the manuscript was well-written, and the results logically follow from their model. However, fundamental questions about the model itself are important to consider. Recently, Buetfering and colleagues found that parvalbumin-positive (PV+) interneurons in the medial entorhinal cortex (MEC) integrate input form grid cells with various phases, and exhibited low spatial sparsity and no spatial periodicity (*[4]*. Nat Neurosci 17(5):710–718.). This argues against a scenario in which fast-spiking interneurons mediate grid cell phase-dependent recurrent inhibition in the MEC – a necessary component of the authors' recurrent attractor network model, where the activity bump in the E-cell population is reflected by an inverted bump in I-cell population activity. This discrepancy needs to be addressed*.

We appreciate the reviewers point concerning comparisons with the Buetfering et al. dataset, but several issues are important to note.

1) Our new analysis of I cell firing fields indicates that the Buetfering et al. data are in fact consistent with the predictions of the E-I. Importantly, we show that to be consistent with the models we consider here the experimentally observed interneuron firing fields need not have high spatial periodicity. This is in contrast to the assumption made by Buetfering et al. when interpreting their data. These issues are outlined in more detail in our response to Reviewer 1 above.

2) Our new simulations, described above in response to Reviewer 2, extend our observations to networks in which E–E coupling is present. They show that our general conclusions hold in these networks too. They also argue that if grid fields are generated through attractor network mechanisms, for which there is now considerable experimental support (e.g. see [48]; Domnisorou et al., 2013; Schmidt-Hieber et al., 2014), then interneurons with spatial firing fields, although not necessarily rotational symmetry of their spatial autocorrelations, are likely to be present regardless of the exact underlying mechanism.

3) The E-I model does not argue that specifically parvalbumin positive interneurons mediate E-I connectivity underlying grid firing. Thus, other interneuron subtypes not investigated by Buetfering et al. are equally plausible candidates.

While our previous manuscript overlooked these issues, in the revised manuscript we now make sure they are clearly discussed. Importantly, all of the models we now consider (E-I, E-E-I and E-I-I) support our general conclusions regarding the relationships between noise, g_E_ and g_I_.

[Editors’ note: the author responses to the re-review follow.]

Reviewer #1 Minor Comments:

*I have one minor question, in their new*
Figure 2*, they show unimodal spatial autocorrelation images of the I cells. This does occur in the Buetfering paper*
Figure 4*, but there are also a few with a weak periodic pattern (lower right panel of*
Figure 4*, grid score 0.18). Can this be found in the simulations as well*?

While our new simulations (Figure 3–figure supplement 5) establish in principle that E-I based attractor networks can generate grid outputs even when spatial firing of many E and I cells in the network is not clearly grid-like, the extent to which these networks can account for all details of experimental observations, including the weak periodic pattern referred to by the reviewer, is not yet clear. We have modified the Discussion to highlight this point.

Reviewer #3:

*Overall, we think that the authors have addressed our concerns. In particular, the additional simulations and discussions about how their findings relate to*
[4]*, examining the effect of recurrent inhibition (I–I) has made the paper much stronger, and tying their results into some experimentally testable predictions*.

*The general results showing how different mechanisms could control grid activity and gamma oscillations are interesting although a bit more of an explanation of the precise mechanisms of their gamma oscillation is needed (i.e., first paragraph of Discussion)*.

We have modified the second sentence of the first paragraph of the Discussion to try to address this point.